# Diffusion Preference Alignment via Relative Text-Image Contrast

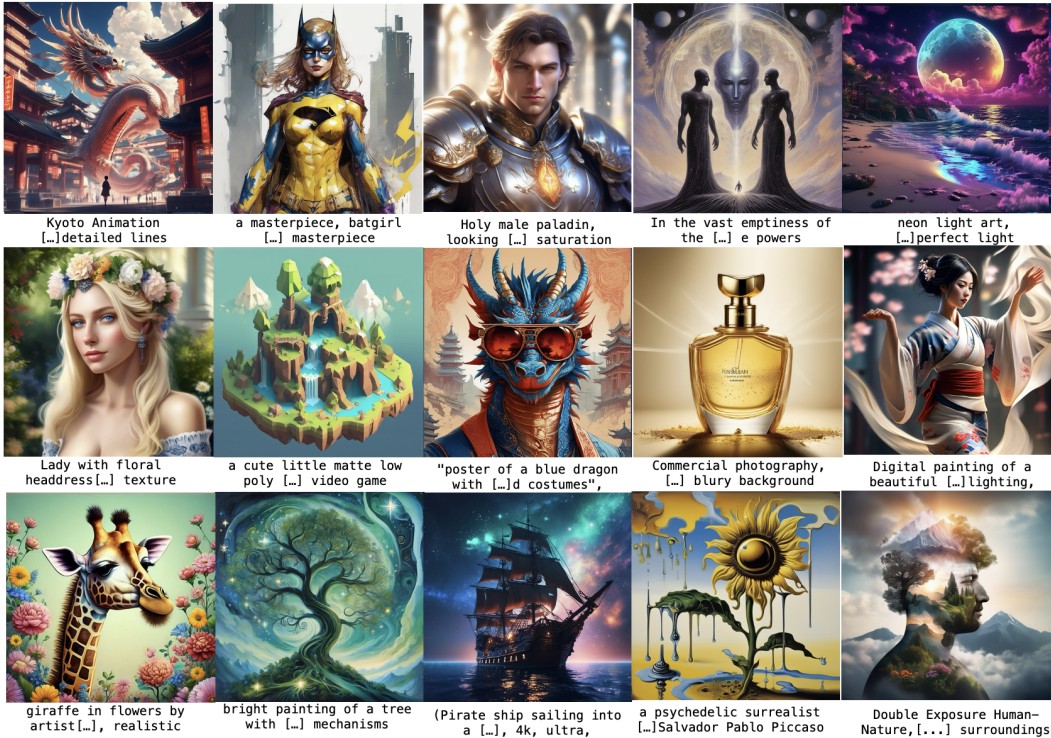

Figure 1: **Diff-contrast represents a novel approach that aligns Text-to-Image models with human preferences by optimizing diffusion model sampling steps and applying contrastive weighting to similar prompt-image pairs.** As demonstrated by the samples above, the Diff-contrast fine-tuned SDXL-1.0 model successfully generates images that closely align with human preferences. The list of prompts is provided in the Appendix.

## Abstract

Aligning large language models with human preferences has emerged as a critical focus in language modeling research. Yet, integrating preference learning into Text-to-Image (T2I) generative models is still relatively uncharted territory. The Diffusion-DPO technique made initial strides by employing pairwise preference learning in diffusion models tailored for specific text prompts. However, Diffusion-DPO overlooks the valuable information that can be derived from the contrast between images with related prompts, which could further enhance the quality of text-to-image (T2I) generation. We introduce Diff-contrast, a new method designed to align diffusion-based T2I models with human preferences more effectively by optimizing relative preferences. This approach leverages both prompt-image pairs with identical prompts and those with semantically related content across various modalities. Furthermore, we have developed a new evaluation metric, style alignment, aimed at overcoming the challenges of high costs, low reproducibility, and limited interpretability prevalent in current evaluations of human preference alignment. Our findings demonstrate that Diff-contrast outper-

forms established methods such as Supervised Fine-Tuning and Diffusion-DPO in tuning Stable Diffusion versions 1.5 and XL-1.0, achieving superior results in both automated evaluations of human preferences and style alignment.

# 1 INTRODUCTION

Diffusion-based Text-to-Image (T2I) models have become the gold standard in image generative technologies. These models are typically pre-trained on extensive web datasets. However, this one-stage training approach may generate images that do not align with human preferences. In contrast, Large Language Models (LLMs) have seen significant advancements in producing outputs that cater to human preferences, primarily through a two-stage training process involving pre-training on web datasets followed by fine-tuning on preference data (Rafailov et al., 2023; Yin et al., 2024; Chen et al., 2024). Extending this preference fine-tuning approach to T2I models (Wallace et al., 2024) presents opportunities to tailor image generation models to cater to diverse user preferences, enhancing their utility and relevance.

Recent research has concentrated on refining diffusion-based T2I models to better reflect human preferences, such as aesthetics and text-image alignment, through Reinforcement Learning from Human Feedback (RLHF) (Black et al., 2024; Clark et al., 2024; Fan et al., 2024; Lee et al., 2023; Prabhudesai et al., 2023; Xu et al., 2024). This process typically involves pretraining a reward model to represent specific human preferences and then optimizing the diffusion models to maximize the reward of generated images. However, developing a robust reward model that accurately mirrors human preferences is challenging and computationally expensive. Over-optimizing the reward model often leads to significant issues of model collapse (Prabhudesai et al., 2023; Lee et al., 2023).

Diffusion-DPO (Wallace et al., 2024) and incorporates Direct Preference Optimization (DPO) (Rafailov et al., 2023) into the preference learning framework of T2I diffusion models. DPO in LLMs focuses on contrasting chosen and rejected responses, bypassing the need for training additional reward models. However, DPO may not fully capture the nuances of human learning, which benefits from analyzing both successful examples and relevant failures (Dahlin et al., 2018). D3PO Yang et al. (2024) incorporates the DPO framework in online setting by sampling preference pairs from the T2I model. However, the on-the-fly perference generation incurs extra computing cost.

Relative Preference Optimization (RPO) (Yin et al., 2024) proposes a learning method akin to human learning, where insights are derived from contrasting both identical and similar questions. RPO contrasts all chosen and rejected responses within a mini-batch, weighting each pair according to the similarity of their prompts. This contrastive preference learning approach has shown convincing improvements in aligning human preferences in LLMs compared to other baseline methods. Further discussions on related works can be found in Appendix B.

In this paper, we aim to design an alignment algorithm for text-to-image (T2I) models that effectively leverages the information contained in data with non-identical prompts. By contrasting images generated from non-identical prompts, Diff-contrast can help the model discern overarching patterns in color, lighting effects, and composition that align more closely with human preferences (Palmer et al., 2013). However, designing such a contrastive alignment algorithm for T2I diffusion models presents several significant challenges. Firstly, the log density of the final generated images is implicit and challenging to quantify in diffusion models because it involves integrating out all intermediate steps. Additionally, T2I diffusion models are inherently multi-modal, involving input prompts and output images in different modalities, which complicates the measurement of similarity between these multi-modal input-output pairs.

To overcome these challenges, we have 1) derived the Diff-contrast loss for diffusion models, simplifying it to apply relative preference alignment across each timestep, and 2) implemented the CLIP (Radford et al., 2021) encoder to project prompts and images into the same embedding space. This allows for the accurate measurement of similarity between multi-modal prompt-image pairs. Our experiments, as illustrated in Figure 2, validate our approach, demonstrating that learning preferences across non-identical prompts significantly enhances the alignment of generated images with human preferences in T2I models.

Moreover, our experiments highlight several shortcomings in the current evaluation metrics for human preference alignment. Traditional approaches like Diffusion-DPO rely on human evaluators,

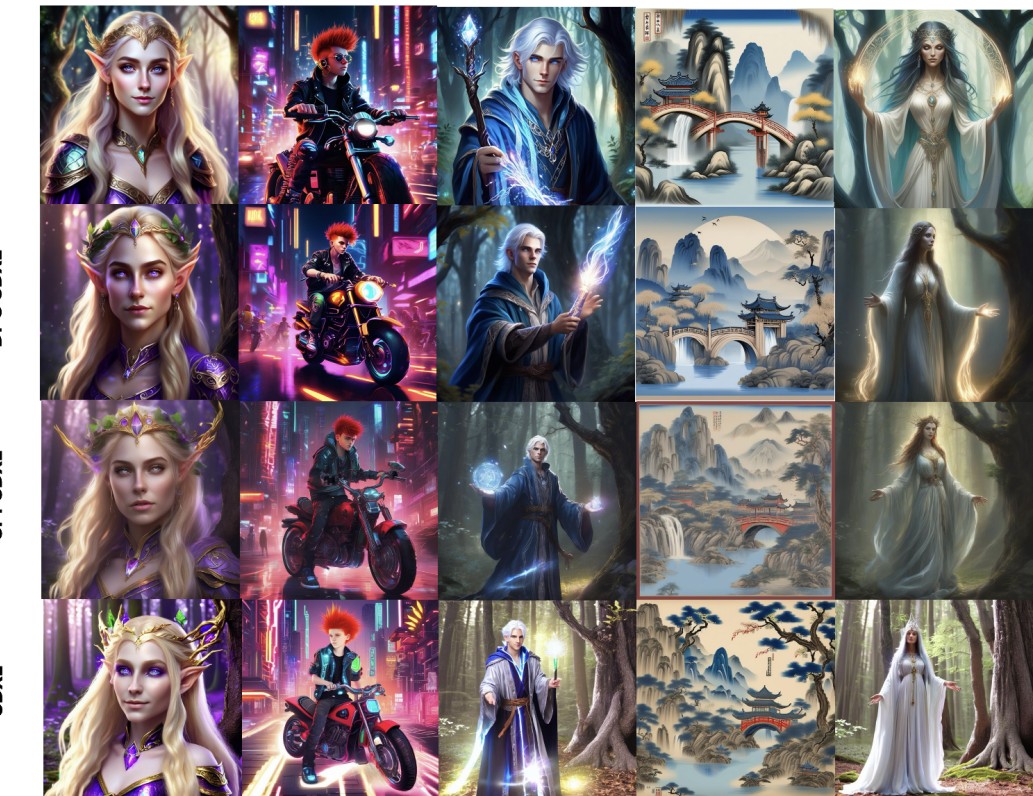

Figure 2: **Sample images from Diff-contrast-SDXL** The prompts used to generate the images are: "A fantasy-themed portrait of a female elf with golden hair and violet eyes, her attire shimmering with iridescent colors, set in an enchanted forest. 8K, best quality, fine details.", "A rebellious teenage boy with spiked, vibrant red hair, riding a futuristic motorcycle through neon-lit city streets, headphones around his neck blaring electronic music. Best quality, fine details", "A young male mage with silver hair and mysterious blue eyes, wearing an intricate robe, casting spells with a glowing crystal staff in a dark enchanted forest. Best quality, fine details.", and "The image features an ancient Chinese landscape with a mountain, waterfalls, willow trees, and arch bridges set against a blue background.","A mysterious priestess in a flowing, ethereal gown, communicating with spirits in a sacred grove, her eyes glowing when she invokes ancient spells. Best quality, fine details."

which not only incur substantial costs but also lead to results that are difficult to reproduce. While various reward models, such as HPSv2 (Wu et al., 2023), Pick Score (Kirstain et al., 2023), and ImageReward (Xu et al., 2024), have been pretrained on datasets labeled by humans to represent human preferences, the minimal variance in their reward scores often understates the differences in images as perceived by human preferences, complicating the assessment of whether a preference learning algorithm genuinely enhances alignment, as illustrated in Table 1.

To more effectively and reliably evaluate preference learning algorithms, we introduce a novel task called Style Alignment and have developed dedicated datasets for this purpose. This task aims to align the outputs of T2I models with specific styles, such as Van Gogh or sketch, identified as preferred samples within the dataset. We then measure the success of preference learning methods based on how closely the images generated by the fine-tuned model match these preferred styles.

We conduct empirical evaluations of Diff-contrast on state-of-the-art T2I models, including Stable Diffusion 1.5 (Rombach et al., 2022a) (SD1.5) and XL-1.0 (Podell et al., 2023b) (SDXL), and compare it with leading image preference alignment methods. Our experimental results show that Diff-contrast outperforms baseline methods in both human preference alignment and style alignment tasks by clear margins.

Our main contributions are summarized as follows:

- We introduce a simplified step-wise denoising alignment loss and multi-modal re-weighting factors for improved effectiveness.

- We introduce Style Alignment, a new evaluation task for image preference learning that is less costly than using human labelers and yields more reproducible and interpretable results compared to traditional human preference reward models.

- Diff-contrast outperforms existing preference learning baselines across both automated evaluations of human preference and style alignment tasks.

## 2 DIFF-CONTRAST

In this section, we introduce Diff-contrast, a new diffusion-based preference learning algorithm that addresses the two challenges mentioned above. In LLM, $\pi_\theta(\boldsymbol{y} \mid \boldsymbol{x})$ is readily estimated through next-token prediction, given a specific response $\boldsymbol{y}$ to a prompt $\boldsymbol{x}$. However, diffusion models approximate the probability of generating an image by reversing the forward diffusion process, $\pi_\theta(\boldsymbol{y}_{0:T} \mid \boldsymbol{x}) = \pi(\boldsymbol{y}_T) \prod_{t=T-1}^{t=1} \pi_\theta(\boldsymbol{y}_t \mid \boldsymbol{y}_{t+1}, \boldsymbol{x})$. To ease notation, we denote $\pi_\theta(\boldsymbol{y}_t \mid \boldsymbol{y}_{t+1}, \boldsymbol{x})$ as $\pi_\theta(\boldsymbol{y}_t \mid \boldsymbol{y}_{t+1})$ when there is no ambiguity. Integrating out all intermediate steps $\boldsymbol{y}_t \mid \boldsymbol{y}_{t+1}$ for all $t < T$ is not feasible in practice due to the huge computational cost. Hence, we propose Diff-contrast and perform preference optimization at each diffusion reversing step. Following Ethayarajh et al. (2024) and Yin et al. (2024), we define the step-wise reward function for diffusion models as

$$r(\boldsymbol{y}_t^w \mid \boldsymbol{y}_{t+1}^w) = \beta \log \frac{\pi_\theta(\boldsymbol{y}_t^w \mid \boldsymbol{y}_{t+1}^w)}{\pi_{\text{ref}}(\boldsymbol{y}_t^w \mid \boldsymbol{y}_{t+1}^w)} \tag{1}$$

where $\beta$ is the regularization hyper-parameter as in Wallace et al. (2024) and Rafailov et al. (2023). Then, we define the Diff-contrast loss at each reverse diffusion step $t$ as:

$$\mathcal{L}_{\text{Diff-contrast},t}(\theta) = \omega_{i,j} \log \sigma \left( \beta \log \frac{\pi_\theta(\boldsymbol{y}_{t,i}^w \mid \boldsymbol{y}_{t+1,i}^w)}{\pi_{\text{ref}}(\boldsymbol{y}_{t,i}^w \mid \boldsymbol{y}_{t+1,i}^w)} - \beta \log \frac{\pi_\theta(\boldsymbol{y}_{t,j}^l \mid \boldsymbol{y}_{t+1,j}^l)}{\pi_{\text{ref}}(\boldsymbol{y}_{t,j}^l \mid \boldsymbol{y}_{t+1,j}^l)} \right) \tag{2}$$

Note that different from Yin et al. (2024), the embedding distance weighting is performed outside the log-sigmoid function as we have observed superior empirical performance with this modified formulation.

Following DDPM (Ho et al., 2020), we then optimize the diffusion models across all timesteps $t$,

$$\tilde{\mathcal{L}}(\theta) = \mathbb{E}_{t \sim \mathcal{U}[1,T-1]} \left[ \lambda(t) \cdot \omega_{i,j} \log \sigma \left( \beta \log \frac{\pi_\theta(\boldsymbol{y}_{t,i}^w \mid \boldsymbol{y}_{t+1,i}^w)}{\pi_{\text{ref}}(\boldsymbol{y}_{t,i}^w \mid \boldsymbol{y}_{t+1,i}^w)} - \beta \log \frac{\pi_\theta(\boldsymbol{y}_{t,j}^l \mid \boldsymbol{y}_{t+1,j}^l)}{\pi_{\text{ref}}(\boldsymbol{y}_{t,j}^l \mid \boldsymbol{y}_{t+1,j}^l)} \right) \right] \tag{3}$$

where $t$ is uniformly sampled from $[1, T-1]$, and $\lambda(t)$ is the timestep re-weighting factor. Similar to Ho et al. (2020), this re-weighting factor is chosen to be constant for all timesteps. In the following two sections, we will explore **(a)** how to define $\omega_{i,j}$ for multi-modal T2I tasks and **(b)** how to efficiently estimate $\pi_\theta(\boldsymbol{y}_t^l \mid \boldsymbol{y}_{t+1}^l)$ by using only $(\boldsymbol{y}_0^w, \boldsymbol{y}_0^l, \boldsymbol{x})$ from the offline dataset.

### 2.1 CONTRASTIVE WEIGHTS FOR IMAGE PREFERENCE LEARNING

In RPO (Yin et al., 2024), the weight $\omega_{i,j}$ is assigned to a pair of preference data, denoted as $(\boldsymbol{y}_i^w, \boldsymbol{x}_i^w)$ and $(\boldsymbol{y}_j^l, \boldsymbol{x}_j^l)$, by comparing their text embeddings. This weighting scheme utilizes the semantically related preference pairs within the batch and is the core to the success of RPO. In T2I models, we have multi-modal preference data with prompt and image. To this end, we introduce multi-modal distance weights that consider the context of both the prompt and image. This adaptation is designed to enhance the learning of preferences in T2I models by capturing the intricate interplay between textual and visual modalities.

**Multi-Modal Embedding Distance Weights** Denote $(\boldsymbol{y}, \boldsymbol{x})$, $(\boldsymbol{y}', \boldsymbol{x}')$ as two image-prompt pairs and $f(\boldsymbol{y}, \boldsymbol{c})$ as an encoder that extracts image-text embeddings. The contra distance weight is defined as

$$\tilde{\omega} = \exp(-\frac{\cos(f(\boldsymbol{y}, \boldsymbol{x}), f(\boldsymbol{y}', \boldsymbol{x}'))}{\tau})$$

where $\cos(f(\boldsymbol{y}, \boldsymbol{x}), f(\boldsymbol{y}', \boldsymbol{x}')) = 1 - \frac{f(\boldsymbol{y},\boldsymbol{x})^T f(\boldsymbol{y}',\boldsymbol{x}')}{||f(\boldsymbol{y},\boldsymbol{x})||_2||f(\boldsymbol{y}',\boldsymbol{x}')||_2}$, $\tau$ is the temperature parameter that determines the sensitivity of the weight against the change in embedding distance. Smaller $\tau$ leads to greater sensitivity to variations in embedding distance, resulting in weights that focus on pairs that are very close in terms of semantic meaning and visual appearance. Conversely, a higher $\tau$ leads to more uniform weight distribution.

In our setting with paired preference data: $\mathcal{D} = (\boldsymbol{y}^w, \boldsymbol{y}^l, \boldsymbol{x})$. Let $M$ be the size of the mini-batch. The unnormalized distance weight matrix $\tilde{\mathbf{W}} \in \mathbb{R}^{M \times M}$ is then defined as $\tilde{\mathbf{W}}_{i,j} = \tilde{\omega}_{i,j}$, where $\tilde{\omega}_{i,j} = \exp(-\frac{\cos(f(\boldsymbol{y}_i^w, \boldsymbol{x}_i^w), f(\boldsymbol{y}_j^l, \boldsymbol{x}_j^l))}{\tau})$. Following Yin et al. (2024), we normalize the distance weights on each row of the weight matrix so that the sum of each row of $\mathbf{W}$ equals 1. Intuitively, the distance matrix measures the similarity between preference pairs $(\boldsymbol{y}_i^w, \boldsymbol{x}_i^w)$ and $(\boldsymbol{y}_j^l, \boldsymbol{x}_j^l)$, $i, j \in [M]$. The diagonal elements corresponds to the preference pairs with identical prompt. The off-diagonal elements represent the contrast within the batch among preference pairs associated with different prompts. This approach leverages all available preference pairs within the batch and enhances image preference learning.

## 2.2 Sampling Diffusion Chain from Offline data

Given $(\boldsymbol{y}_0^w, \boldsymbol{y}_0^l, \boldsymbol{x})$, Diff-contrast loss in Equation (3) requires samples $\boldsymbol{y}_t, \boldsymbol{y}_{t+1}$ to evaluate the log probability $\log \pi_\theta(\boldsymbol{y}_t \mid \boldsymbol{y}_{t+1})$. Note that $\boldsymbol{y}_t, \boldsymbol{y}_{t+1}$ are dependent with respect to the same $\boldsymbol{y}_0$, and one can be efficiently derived from the forward process of Diffusion-based generative models (Ho et al., 2020). The forward process is defined to gradually add Gaussian noise to the data $\boldsymbol{y}_0 \sim q(\boldsymbol{y}_0)$ according to a variance schedule $\beta_1, \ldots, \beta_T$:

$$q\left(\boldsymbol{y}_{1:T} \mid \boldsymbol{y}_0\right) := \prod_{t=1}^T q\left(\boldsymbol{y}_t \mid \boldsymbol{y}_{t-1}\right), \quad q\left(\boldsymbol{y}_t \mid \boldsymbol{y}_{t-1}\right) := \mathcal{N}\left(\boldsymbol{y}_t; \sqrt{1 - \beta_t}\boldsymbol{y}_{t-1}, \beta_t\boldsymbol{I}\right) \quad (4)$$

Next, we introduce how we derive $\boldsymbol{y}_t$, $\boldsymbol{y}_{t+1}$ and $\pi_\theta(\boldsymbol{y}_t \mid \boldsymbol{y}_{t+1})$.

**Maginal Sampling**  The forward process admits sampling $\boldsymbol{y}_t$ at an arbitrary timestep $t$ in closed form. Let $\alpha_t := 1 - \beta_t$ and $\bar{\alpha}_t := \prod_{s=1}^t \alpha_s$. With the re-parametrization trick, we have

$$\boldsymbol{y}_{t+1} = \sqrt{\bar{\alpha}_t}\boldsymbol{y}_0 + (1 - \bar{\alpha}_t)\boldsymbol{\epsilon}_{t+1}, \boldsymbol{\epsilon}_{t+1} \sim \mathcal{N}(\boldsymbol{0}, \boldsymbol{I}) \quad (5)$$

For $\forall t$, we directly sample $\boldsymbol{y}_{t+1}$ according to Equation (5) in the forward diffusion process.

**Denoising with Ground Truth**  Following the usual practice (Ho et al., 2020; Xiao et al., 2021; Wang et al., 2023b; Zhou et al., 2023), we sample $\boldsymbol{y}_t|\boldsymbol{y}_{t+1}, \boldsymbol{y}_0$ from the conditional posterior distribution of previous diffusion steps,

$$q\left(\boldsymbol{y}_t \mid \boldsymbol{y}_{t+1}, \boldsymbol{y}_0\right) = \mathcal{N}\left(\boldsymbol{y}_t; \tilde{\boldsymbol{\mu}}_{t+1}\left(\boldsymbol{y}_{t+1}, \boldsymbol{y}_0\right), \tilde{\beta}_{t+1}\boldsymbol{I}\right), \quad (6)$$

where $\tilde{\boldsymbol{\mu}}_{t+1}\left(\boldsymbol{y}_{t+1}, \boldsymbol{y}_0\right) := \frac{\sqrt{\bar{\alpha}_t}\beta_{t+1}}{1 - \bar{\alpha}_{t+1}}\boldsymbol{y}_0 + \frac{\sqrt{\alpha_{t+1}}(1 - \bar{\alpha}_t)}{1 - \bar{\alpha}_{t+1}}\boldsymbol{y}_{t+1}$ and $\tilde{\beta}_{t+1} := \frac{1 - \bar{\alpha}_t}{1 - \bar{\alpha}_{t+1}}\beta_t$. Using the Gaussian re-parametrization, and replacing the $\boldsymbol{y}_0$ with $\boldsymbol{y}_{t+1}$ in Equation (5), we could obtain

$$\boldsymbol{y}_t = \sqrt{\frac{\alpha_t}{\alpha_{t+1}}}(\boldsymbol{y}_{t+1} - \frac{\beta_{t+1}}{\sqrt{1 - \bar{\alpha}_{t+1}}}\boldsymbol{\epsilon}_{t+1}) + \sigma_{t+1}\boldsymbol{\epsilon}_t$$

where $\sigma_{t+1}^2 = \tilde{\beta}_{t+1} = \frac{1 - \bar{\alpha}_{t1}}{1 - \bar{\alpha}_{t+1}}\beta_{t+1}$, $\epsilon_t \sim \mathcal{N}(0, \boldsymbol{I})$. Following the denoising model definition in DDPM (Ho et al., 2020), we parameterize $\pi_\theta(\boldsymbol{y}_t \mid \boldsymbol{y}_{t+1})$ as follows :

$$\pi_\theta(\boldsymbol{y}_t \mid \boldsymbol{y}_{t+1}) = \mathcal{N}(\boldsymbol{y}_t; \sqrt{\frac{\alpha_t}{\alpha_{t+1}}}(\boldsymbol{y}_{t+1} - \frac{\beta_{t+1}}{\sqrt{1 - \bar{\alpha}_{t+1}}}\boldsymbol{\epsilon}_\theta(\boldsymbol{y}_{t+1}, t+1)), \sigma_{t+1}^2\boldsymbol{I}) \quad (7)$$

For simplicity, we approximate $\boldsymbol{y}_t$ with its posterior mean: $\mathbb{E}\left[\boldsymbol{y}_t \mid \boldsymbol{y}_{t+1}, \boldsymbol{y}_0\right] = \sqrt{\frac{\alpha_t}{\alpha_{t+1}}}(\boldsymbol{y}_{t+1} - \frac{\beta_{t+1}}{\sqrt{1 - \bar{\alpha}_{t+1}}}\boldsymbol{\epsilon}_{t+1})$. We then plug $\boldsymbol{y}_t$ and $\boldsymbol{y}_{t+1}$ into Equation (7), and derive

$$\pi_\theta(\boldsymbol{y}_t|\boldsymbol{y}_{t+1}) \approx \frac{1}{\left(\sqrt{2\pi\sigma_{t+1}^2}\right)^d} \exp\left(-\frac{1}{2}\frac{\beta_{t+1}}{(1 - \bar{\alpha}_t)}\frac{\alpha_t}{\alpha_{t+1}}\left\|\boldsymbol{\epsilon}_\theta(\boldsymbol{y}_{t+1}, t+1) - \boldsymbol{\epsilon}_{t+1}\right\|_2^2\right) \quad (8)$$

where $d$ is the dimension of the image vector.

Note that $\pi_\theta(\boldsymbol{y}_t|\boldsymbol{y}_{t+1})$ follows the same formula for both preferred and rejected sample pairs. Combining Equations (3) and (8), with a constant timestep re-weighting (Ho et al., 2020; Wallace et al., 2024), yields our final Diff-contrast loss:

$$
\mathcal{L}_{\text{Diff-contrast}}(\theta) = -\mathbb{E}\Bigg[\omega_{i,j}\log\sigma\Bigg(-\frac{1}{2}\beta\big(\big\|\boldsymbol{\epsilon}_\theta(\boldsymbol{y}_{t+1,i}^w,t)-\boldsymbol{\epsilon}^w\big\|_2^2 - \big\|\boldsymbol{\epsilon}_{\text{ref}}(\boldsymbol{y}_{t+1,i}^w,t)-\boldsymbol{\epsilon}^w\big\|_2^2
$$
$$
-\left(\big\|\boldsymbol{\epsilon}_\theta(\boldsymbol{y}_{t+1,j}^l,t)-\boldsymbol{\epsilon}^l\big\|_2^2 - \big\|\boldsymbol{\epsilon}_{\text{ref}}(\boldsymbol{y}_{t+1,j}^l,t)-\boldsymbol{\epsilon}^l\big\|_2^2\right)\big)\Bigg)\Bigg]
\tag{9}
$$

We provide the detailed derivation in Appendix A.

We notice a strong connection between the loss shown above with that of conditional transport (Zheng & Zhou, 2021; Tanwisuth et al., 2021), where $\omega_{ij}$ with $\sum_j \omega_{ij} = 1$ can be interpreted as the probability of transporting to the $j$th losing prompt-image pair $(\boldsymbol{x}_j, \boldsymbol{y}_{t+1,j}^l)$ given the $i$th winning prompt-image pair $(\boldsymbol{x}_i, \boldsymbol{y}_{t+1,i}^w)$ as the origin of the transport, while the $(i, j)$ term transformed by $\log\sigma(\cdot)$ would be considered as the cost of transporting from $(\boldsymbol{x}_i, \boldsymbol{y}_{t+1,i}^w)$ to $(\boldsymbol{x}_j, \boldsymbol{y}_{t+1,j}^l)$. From the conditional transport perspective, for each winning pair $(\boldsymbol{x}_i, \boldsymbol{y}_{t+1,i}^w)$, Diff-contrast computes the distance between the winning pair and all losing pairs in the mini-batch and prioritizes maximizing the cost from transporting from $(\boldsymbol{x}_i, \boldsymbol{y}_{t+1,i}^w)$ to the losing pairs in CLIP latent space. This approach effectively minimizes the likelihood of sampling losing images that are semantically related to $(\boldsymbol{x}_i, \boldsymbol{y}_{t+1,i}^w)$. This perspective may allow further improvement by adhering to the conditional-transport framework, yet modifying the definitions of transport probabilities and point-to-point transport cost to better serve the purpose of preference optimization. We leave this investigation to future study.

## 3  STYLE ALIGNMENT DATASET

The state-of-the-art evaluation metric for human preference learning on T2I models consists of two components: (a) Human evaluation via employed labelers, (b) Automatic evaluation leveraging human preference reward models like HPSV2 (Wu et al., 2023) and Pick Score (Kirstain et al., 2023). (1) **Cost and Reproducibility in Human Evaluation:** Diffusion-DPO (Wallace et al., 2024) evaluates employed labelers on Amazon Mechanical Turk to compare image generations which is costly and suffers from poor reproducibility due to the subjective nature of human preferences. (2) **Difficulties in Automatic Evaluation:** As suggested by Wallace et al. (2024), diffusion-based T2I models are pre-trained on datasets designed to align with human preferences. Consequently, Vanilla T2I models are inherently task-tuned towards human preference alignment. This scenario presents two significant challenges:

(a) Preference learning shows limited improvement in task-tuned models, a phenomenon also observed in the LLM summarization task by Rafailov et al. (2023). (b) The performance of Supervised Fine-Tuning (SFT) varies depending on the capacity of the Vanilla model selected. Both the pre-training and fine-tuning data are collected to enhance alignment with human preferences, making the quality of the preference dataset pivotal to SFT performance. SFT can dramatically improve evaluation scores when the preference data is of higher quality than the model's generations. Conversely, it can degrade performance if the model was pre-trained on images of superior quality. This variability can undermine the effectiveness of automatic evaluations.

These issues highlight the complex interplay between model capacity and data quality of fine-tuning models that are already implicitly preference-tuned during pre-training.

To address the limitations inherent in current image preference evaluation metrics, we introduce a novel downstream task with datasets specifically designed for image preference learning, which we term **Style Alignment**. Style alignment aims to fine-tune T2I models to generate images that align with the offline data to achieve style transfer (Gatys et al., 2015; 2016) on T2I models. These styles are intentionally chosen to be visually distinct from those in the Pick-a-Pic V2 dataset, thereby enhancing the divergence between downstream tasks and pretraining. We constructed three datasets in Van Gogh, Sketch, and Winter styles each containing 10,000 preference pairs.

**Preference Pair Construction**    The concept of style transfer is particularly well-suited to the field of image preference learning. We label the edited image after style transfer as preferred and the original image as rejected to fine-tune T2I models towards generating images with desired style. Our datasets are crafted based on subsets of Pick-a-Pic V2 (Kirstain et al., 2023). For each pair of images in the subset, we randomly pick one as rejected and process it with state-of-the-art image editing models: Prompt Diffusion (Wang et al., 2023a) for Sketch and Winter style and Instruct Pix2Pix (Brooks et al., 2023) for Van Gogh style. The new image after style transfer is labeled as preferred. Notationally, the new datasets follow the same format as Pick-a-Pic V2: $(\boldsymbol{y}^w, \boldsymbol{y}^l, \boldsymbol{x})$, where $\boldsymbol{y}^w$ is the image after style transfer, $\boldsymbol{y}^l$ is the image from Pick-a-Pic.

**Content-Safer Dataset**    Pick-a-Pic V2 is collected through webapp without safety checking. We observed substantial amount of inappropriate content when exploring the dataset. To foster a safe and ethical research environment, we manually deleted the inappropriate prompts when creating our style transfer datasets. We will release our datasets to the public soon.

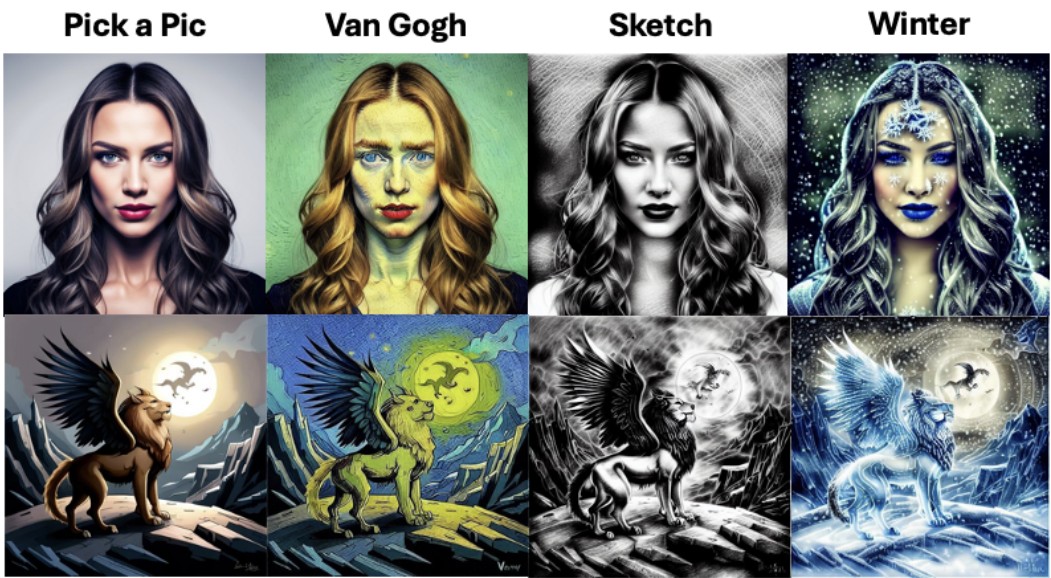

Figure 3: **Example Images from Pick-a-pic, Van Gogh, Sketch and Winter Datasets**

## 4    EXPERIMENTS

In our study, we conduct experiments designed to address the following research questions: (1) Can Diff-contrast enhance the T2I models' ability to generate images that better align with human preference? (2) How do preference learning methods perform when applied to style alignment tasks? (3) What are the key factors that affect the performance of Diff-contrast? (4) Could style alignment effectively address the challenges encountered in aligning with human preferences in image preference learning evalutaions?

### 4.1    EXPERIMENT SETUP

**Models and Datasets**    Following Wallace et al. (2024), our human preference alignment experiments are conducted on the Pick-a-Pic V2 dataset (Kirstain et al., 2023). We conduct style alignment experiments on our style alignment datasets (Van Gogh, Sketch, and Winter). The experiments are conducted on Stable Diffusion 1.5 (SD1.5) and Stable Diffusion XL (SDXL).

**Evaluation**    For human preference alignment, we evaluated the Pick Score, HPSV2 , LAION Aesthetics score (Schuhmann, 2022), CLIP (Radford et al., 2021), and ImageReward (Xu et al., 2024), using the HPSV2 benchmark test prompts (Wu et al., 2023) and Parti-Prompts (Yu et al., 2022).

With each reward model, we report both average scores and win rates between Diff-contrast and the baselines. For style alignment, we quantified model performance using the Fréchet Inception Distance (FID) (Heusel et al., 2017), which measures the difference between the preferred images in our training set and those generated from the training prompts by the fine-tuned T2I model.

**Implementations**  We build Diff-contrast upon the codebase of Diffusion-DPO (Wallace et al., 2024). We leverage the CLIP encoder (Radford et al., 2021) to compute multi-modal embeddings. For human preference alignment, we performed one-stage fine-tuning on both models for SFT and Diff-contrast. We directly use the checkpoints provided by Diffusion-DPO[1]. For style alignment, we also conduct one-stage fine-tuning. Additionally, following Rafailov et al. (2023), we perform two-stage fine-tuning which is expected to be more effective due to the domain gap between the pre-training of T2I models and the style alignment task. The details on learning rate and optimization steps are listed in Table 4.

## 4.2 Human Preference Alignment

Table 1 reports win rates for Diff-contrast, Diffusion-DPO (Wallace et al., 2024), SFT, and Base Stable Diffusion. On SDXL, Diff-contrast achieves better performance on all reward models comparing to the baselines except for slightly lower CLIP score on Parti-Prompt dataset. In particular, Diff-contrast achieves **64.28** HPSV2 win rate on the HPSV2 benchmark dataset and **64.09** HPSV2 win rate on the Parti-Prompt dataset . On SD1.5, We observe that Diff-contrast outperforms Diffusion-DPO on HPSV2, PickScore, Aesthtics, and Image Reward. The consistent improvement comparing to Diffusion-DPO (Wallace et al., 2024) in aligning SD1.5 and SDXL showcases that utilizing information from related prompt-image pairs would benefit image preference learning.

Table 1: Win rates for Diff-contrast v.s. existing alignment approaches under various reward models using prompts from the HPSV2 and Parti-Prompt datasets. On SD1.5, the temperature parameter is chosen to be 0.01, on SD1.5 and 0.5 on SDXL

| Model | Test Dataset | Method | HPS | Pick Score | Aesthetics | CLIP | Image Reward |
|---|---|---|---|---|---|---|---|
| SD1.5 | HPSV2 | Contrast (ours) v.s. Base | **88.44** | **88.09** | **78.66** | **58.16** | **80.47** |
| | | Contrast (ours) v.s. SFT | **63.84** | **68.25** | 47.22 | **66.09** | **62.5** |
| | | Contrast (ours) v.s. DPO (Wallace et al., 2024) | **73.41** | **72.28** | **60.19** | 52.09 | **70.03** |
| | Parti-Prompt | Contrast (ours) v.s. Base | **84.19** | **84.87** | **76.65** | **61.34** | **75.06** |
| | | Contrast (ours) v.s. SFT | **71.78** | **75.15** | 47.86 | **65.43** | **64.20** |
| | | Contrast (ours) v.s. DPO (Wallace et al., 2024) | **75.74** | **74.57** | **65.44** | 54.72 | **68.50** |
| SDXL | HPSV2 | Contrast (ours) v.s. Base | **93.09** | **83.59** | **50.59** | **65.28** | **79.41** |
| | | Contrast (ours) v.s. SFT | **79.53** | **95.94** | **81.13** | **62.59** | **78.84** |
| | | Contrast (ours) v.s. DPO (Wallace et al., 2024) | **64.28** | **57.16** | **51.44** | 49.63 | **57.41** |
| | Parti-Prompt | Contrast (ours) v.s. Base | **90.63** | **79.60** | **65.07** | **62.75** | **82.23** |
| | | Contrast (ours) v.s. SFT | **79.78** | **95.10** | **79.41** | **67.40** | **81.00** |
| | | Contrast (ours) v.s. DPO (Wallace et al., 2024) | **64.09** | **58.52** | **59.80** | 48.35 | **58.58** |

Figure 2 provides a visual comparison of Diff-contrast-SDXL, DPO-SDXL, SFT-SDXL, and Base SDXL generations. In general Diff-contrast generates images with better quality. In the first example: "A fantasy-themed portrait of a female elf with golden hair and violet eyes, her attire shimmering with iridescent colors, set in an enchanted forest. 8K, best quality, fine details.", Diff-contrast generation depicts a fantasy-themed elf which is illuminated with vibrant and visually appealing light, highlighting the colorful details and textures of the scene. This kind of lighting creates a lively, enchanting atmosphere. These examples showcase that Diff-contrast is able to generate images with finer details, better lighting effects, vivid colors and higher fidelity to the prompt. More examples from Diff-contrast-SDXL can be found in Appendix F.

## 4.3 Style Alignment

We report the FID between style-transferred images in the training data and the images generated by the finetuned models generated using the same training prompts. The results are provided in Table 2 .

---

[1]The checkpoints are downloaded from https://huggingface.co/mhdang/dpo-sd1.5-text2image-v1 and https://huggingface.co/mhdang/dpo-sdxl-text2image-v1

Table 2: Comparison of the FID scores for images generated from training prompts using style-aligned T2I models, with the training images serving as references.

| Method | SD1.5 | | | SDXL | | |
|---|---|---|---|---|---|---|
| | Van Gogh | Sketch | Winter | Van Gogh | Sketch | Winter |
| SFT | 15.34 | 20.56 | 16.31 | 28.15 | 16.41 | 21.74 |
| DPO (Wallace et al., 2024) | 91.8 | 34.67 | 63.43 | 152.35 | 137.07 | 99.37 |
| Contrast (Ours) | 42.24 | 29.36 | 33.29 | 97.57 | 59.75 | 85.54 |
| SFT+SFT | 14.73 | 19.89 | 16.06 | 22.71 | 30.48 | 30.78 |
| SFT+DPO (Wallace et al., 2024) | 47.47 | 24.92 | 26.26 | 25.62 | 23.32 | 20.02 |
| SFT+Contrast (Ours) | **13.25** | **17.54** | **14.50** | **17.95** | **15.96** | **15.75** |

We observe that for one-stage fine-tuning, SFT is most effective in fine-tuning Vanilla T2I models due to the domain gap between vanilla Stable Diffusion Models and the style alignment dataset. Still, we observe that Diff-contrast better aligns the model to the desired style. In two-stage fine-tuning, Diff-contrast demonstrates superior performance for all 3 styles and 2 models. The outstanding performance of Diff-contrast is attributable to its ability of learning from all comparisons within the mini-batch. The less related prompts-image pairs could still benefit preference learning by the contrast in overall style and backgrounds.

Figure 4 showcases the performance of two-tage style alignment fine-tuning on SD1.5 and SDXL. The terms SFT, DPO, and Diff-contrast atop the image refer to performing SFT, Diffusion-DPO or Diff-contrast on SFT tuned models. We observe that on all three datasets, Diff-contrast successfully learns both details and the overall style, generating images that account for both style characteristics and fine details. we note that Diffusion-DPO tends to learn an incorrect style on the Van Gogh dataset, adding excessive green in the background. On the Winter dataset, Diffusion-DPO tends to over-emphasize the background and obscure the main body's details with snowflakes. Diffusion-DPO encounters similar issues on the Sketch dataset by generating a background that does not resemble a pencil-style sketch and obscures the details with black blocks. SFT can effectively learn the details but appears to be weaker in leaning the styles, especially in the background. We partially attribute the advantage of Diff-contrast over Diffusion-DPO and SFT to the use of related prompt-image pairs. By learning all the pairs in the mini-batch, Diff-contrast is able to capture the styles and composition more accurately, producing better quality images comparing to the baselines. More examples from style aligned Stable Diffusion models can be found in Appendix G.

## 4.4 ABLATION STUDIES

We conducted ablation study on the distance temperature parameter $\tau$ with the aim of understanding how much the focus should we put on semantically related prompt-image pairs to enhance preference alignment. In our experiment, we compare the performance of Diff-contrast under different distance temperatures in both human preference alignment and style alignment. In Table 3, we report reward model scores from automatic evaluations and FID for style alignment. We find that a lower distance temperature ($\tau$ =0.01) leads to overall best performance being highest on HPS (Wu et al., 2023), Image Reward (Xu et al., 2024) and second highest on Pick Score (Kirstain et al., 2023). On the style alignment task, it is observed that the style alignment performance improves as we increase the temperature. This contrast between human preference alignment and style alignment can be attributed to the differing natures of the two tasks: In human preference alignment tasks, a lower temperature focuses more on related prompt-image pairs to improve prompt-specific details. In style alignment tasks, however, correctly learning the overall image style becomes a major challenge. This is tackled by a higher temperature that results in more uniform weights for all preference pairs since any preference pair entails characteristics of image style that contributes to the alignment of styles. Due to page limit, we defer the results to appendix C.

## 5 CONCLUSION

We introduce Diff-contrast as a novel method for aligning Text-to-Image (T2I) Models with human preferences. This approach enhances each sampling step of diffusion models and utilizes similar prompt-image pairs through contrastive weighting. On Stable Diffusion versions 1.5 and XL, Diff-

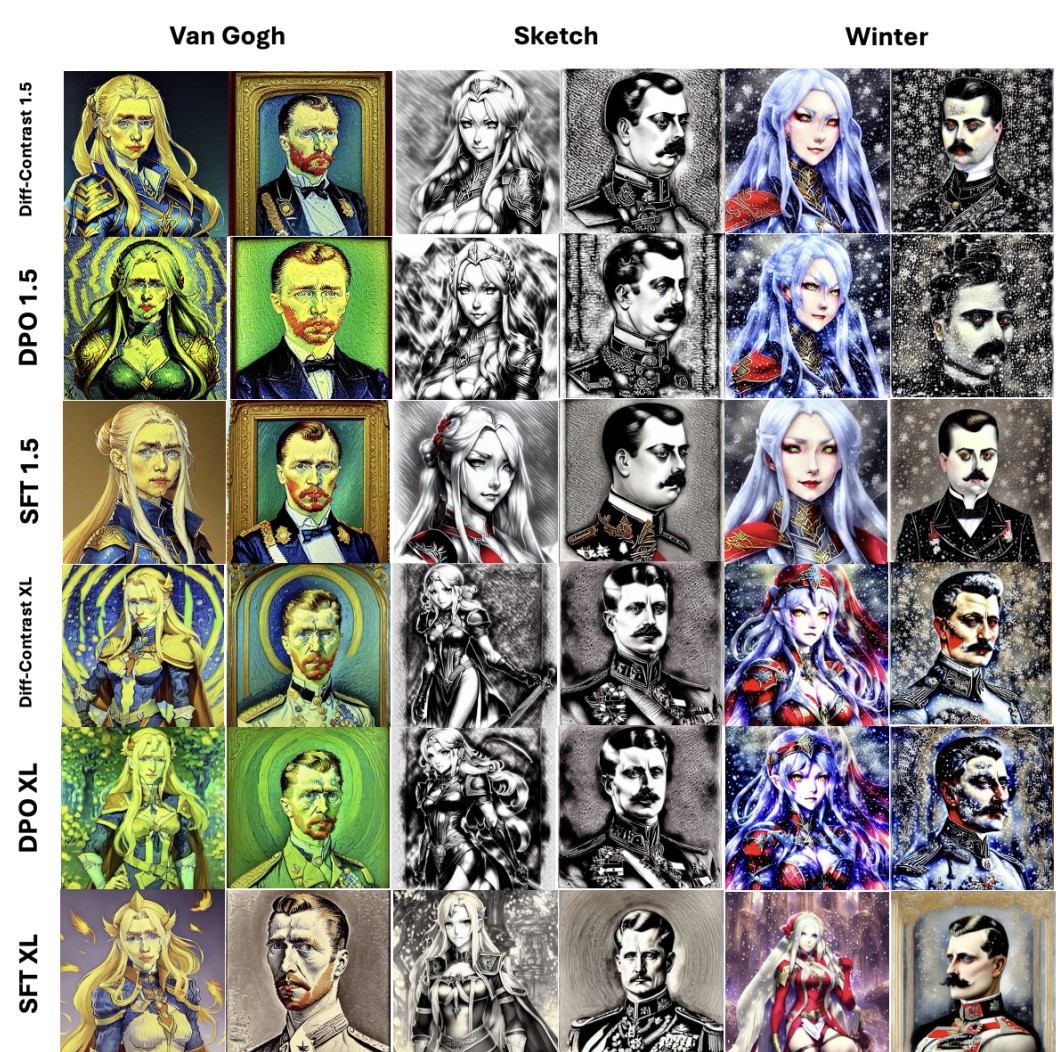

Figure 4: **Sample images from Style Aligned Stable Diffusion Models**, the images are generated from prompts: "Edelgard from Fire Emblem depicted in Artgerm's style.", "Portrait of Archduke Franz Ferdinand by Charlotte Grimm, depicting his detailed face."

contrast exhibited superior performance compared to previous alignment methods in both human preference alignment and our newly introduced style alignment tasks. The empirical findings highlight the effectiveness of Diff-contrast in fine-tuning diffusion-based T2I models and demonstrate the value of style alignment as a reliable measure for assessing image preference learning methods.

**Limitations & Future Work**    Although Diff-contrast enhances the alignment of text-to-image models, it relies on a dataset collected from the web, which may not fully capture the diverse spectrum of human preferences across different communities. A potential avenue for future research could involve developing methods to collect datasets that reflect a wider range of cultural diversity. This would help in creating models that are more universally applicable and sensitive to various cultural nuances.

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

# Diffusion Preference Alignment Via Text- Image Contrast: Appendix

## A MATHEMATICAL DERIVATION

In this section we provide the details on the derivation of our Diff-contrast loss in Equation (9), we start from the loss function in Equation (3):

$$\mathcal{L}_{\text{Diff-contrast},t}(\theta) = \omega_{i,j} \log \sigma \left( \beta \log \frac{\pi_\theta(\boldsymbol{y}_t^w \mid \boldsymbol{y}_{t+1}^w)}{\pi_{\text{ref}}(\boldsymbol{y}_t^w \mid \boldsymbol{y}_{t+1}^w)} - \beta \log \frac{\pi_\theta(\boldsymbol{y}_t^l \mid \boldsymbol{y}_{t+1}^l)}{\pi_{\text{ref}}(\boldsymbol{y}_t^l \mid \boldsymbol{y}_{t+1}^l)} \right) \tag{10}$$

following Ho et al. (2020), the policies are of form:

$$\pi_\theta(\boldsymbol{y}_t^* \mid \boldsymbol{y}_{t+1}^*) = \mathcal{N} \left( \boldsymbol{y}_t^*; \frac{\sqrt{\alpha_t}}{\sqrt{\alpha_{t+1}}} (\boldsymbol{y}_t^* - \frac{\beta_{t+1}}{\sqrt{1 - \bar{\alpha}_{t+1}}} \boldsymbol{\epsilon}_\theta(\boldsymbol{y}_{t+1}^*, t)), \sigma_{t+1}^2 \boldsymbol{I} \right)$$

$$= \frac{1}{\left( \sqrt{2\pi\sigma_{t+1}^2} \right)^d} \exp \left( -\frac{1}{2\sigma_{t+1}^2} \left\| \boldsymbol{y}_t^* - \sqrt{\frac{\alpha_t}{\alpha_{t+1}}} (\boldsymbol{y}_{t+1}^* - \frac{\beta_{t+1}}{\sqrt{1 - \bar{\alpha}_{t+1}}} \boldsymbol{\epsilon}_\theta(\boldsymbol{y}_{t+1}^*, t+1)) \right\|_2^2 \right)$$

where $d$ is the dimension of the image vector, we use $\boldsymbol{y}_t^*$ for ease of notation. The derivation using $\boldsymbol{y}_t^*$ is applicable to both $\boldsymbol{y}_{t,i}^w$ and $\boldsymbol{y}_{t,j}^l$.

The ground-truth denoising distribution and posterior mean can be written in the form:

$$q\left(\boldsymbol{y}_t^* \mid \boldsymbol{y}_t^*, \boldsymbol{y}_0^*\right) = \mathcal{N}(\boldsymbol{y}_t^*; \sqrt{\frac{\alpha_t}{\alpha_{t+1}}} (\boldsymbol{y}_t^* - \frac{\beta_{t+1}}{\sqrt{1 - \bar{\alpha}_{t+1}}} \boldsymbol{\epsilon}_{t+1}), \sigma_{t+1}^2 \boldsymbol{I})$$

$$= \frac{1}{\left( \sqrt{2\pi\sigma_{t+1}^2} \right)^d} \exp \left( -\frac{1}{2\sigma_{t+1}^2} \left\| \boldsymbol{y}_{t+1}^* - \sqrt{\frac{\alpha_t}{\alpha_{t+1}}} (\boldsymbol{y}_{t+1}^* - \frac{\beta_{t+1}}{\sqrt{1 - \bar{\alpha}_{t+1}}} \boldsymbol{\epsilon}_{t+1}) \right\|_2^2 \right)$$

$$\mathbb{E}\left[\boldsymbol{y}_t^* \mid \boldsymbol{y}_{t+1}^*, \boldsymbol{y}_0^*\right] = \sqrt{\frac{\alpha_t}{\alpha_{t+1}}} (\boldsymbol{y}_{t+1}^* - \frac{\beta_{t+1}}{\sqrt{1 - \bar{\alpha}_{t+1}}} \boldsymbol{\epsilon}_{t+1})$$

where $\boldsymbol{y}_0^*$ is from the offline data $\mathcal{D}$, $\boldsymbol{y}_{t+1}^*$ is sampled from the forward process $q(\boldsymbol{y}_{t+1}^* \mid \boldsymbol{y}_0^*)$

If we sample $\boldsymbol{y}_t^*$ from $q\left(\boldsymbol{y}_t^* \mid \boldsymbol{y}_{t+1}^*, \boldsymbol{y}_0^*\right)$, we can express $\boldsymbol{y}_t^* = \sqrt{\frac{\alpha_t}{\alpha_{t+1}}} (\boldsymbol{y}_{t+1}^* - \frac{\beta_{t+1}}{\sqrt{1 - \bar{\alpha}_{t+1}}} \boldsymbol{\epsilon}_{t+1}) + \sigma_{t+1}\boldsymbol{\epsilon}_t$, where $\boldsymbol{\epsilon}_{t+1}$ is the noise added in the forward process to obtain $\boldsymbol{y}_{t+1}^*$, $\boldsymbol{\epsilon}_t$ is the Gaussian noise to obtain $\boldsymbol{y}_t^*$ in the reverse process with the re-parametrization trick. The policy is then evaluated as:

$$\pi_\theta(\boldsymbol{y}_t^*|\boldsymbol{y}_{t+1}^*) = \frac{1}{\left( \sqrt{2\pi\sigma_{t+1}^2} \right)^d} \exp \left( -\frac{1}{2\sigma_t^2} \left\| \sqrt{\frac{\alpha_t}{\alpha_{t+1}}} (\boldsymbol{y}_{t+1}^* - \frac{\beta_{t+1}}{\sqrt{1 - \bar{\alpha}_{t+1}}} \boldsymbol{\epsilon}_{t+1}^*) + \sigma_{t+1}\boldsymbol{\epsilon}_t^* \right.\right.$$

$$\left.\left. - \sqrt{\frac{\alpha_t}{\alpha_{t+1}}} (\boldsymbol{y}_{t+1}^* - \frac{\beta_{t+1}}{\sqrt{1 - \bar{\alpha}_{t+1}}} \boldsymbol{\epsilon}_\theta(\boldsymbol{y}_{t+1}^*, t+1)) \right\|_2^2 \right)$$

$$= \frac{1}{\left( \sqrt{2\pi\sigma_{t+1}^2} \right)^d} \exp \left( -\frac{1}{2\sigma_{t+1}^2} \frac{\alpha_t}{\alpha_{t+1}} \frac{\beta_{t+1}^2}{1 - \bar{\alpha}_{t+1}} \left\| \boldsymbol{\epsilon}_\theta(\boldsymbol{y}_{t+1}^*, t+1) - \boldsymbol{\epsilon}_{t+1}^* + \sigma_{t+1}\boldsymbol{\epsilon}_t^* \right\|_2^2 \right)$$

$$= \frac{1}{\left( \sqrt{2\pi\sigma_{t+1}^2} \right)^d} \exp \left( -\frac{1}{2} \frac{1 - \bar{\alpha}_{t+1}}{(1 - \bar{\alpha}_t)\beta_{t+1}} \frac{\alpha_t}{\alpha_{t+1}} \frac{\beta_{t+1}^2}{1 - \bar{\alpha}_{t+1}} \left\| \boldsymbol{\epsilon}_\theta(\boldsymbol{y}_{t+1}^*, t+1) - \boldsymbol{\epsilon}_{t+1}^* + \sigma_{t+1}\boldsymbol{\epsilon}_t^* \right\|_2^2 \right)$$

$$= \frac{1}{\left( \sqrt{2\pi\sigma_{t+1}^2} \right)^d} \exp \left( -\frac{1}{2} \frac{\beta_{t+1}}{(1 - \bar{\alpha}_t)} \frac{\alpha_t}{\alpha_{t+1}} \left\| \boldsymbol{\epsilon}_\theta(\boldsymbol{y}_{t+1}^*, t+1) - \boldsymbol{\epsilon}_{t+1}^* + \sigma_{t+1}\boldsymbol{\epsilon}_t^* \right\|_2^2 \right)$$

The log probability is simply:

$$\log \pi_\theta(\boldsymbol{y}_t^* | \boldsymbol{y}_{t+1}^*) = -\frac{1}{2}\frac{\beta_{t+1}\alpha_t}{(1-\bar{\alpha}_t)\alpha_{t+1}} \left\| \epsilon_\theta(\boldsymbol{y}_{t+1}^*, t+1) - \epsilon_{t+1}^* + \sigma_{t+1}\epsilon_t^* \right\|_2^2$$
$$-\frac{d}{2}\cdot\log 2\pi - d\cdot\log\sigma_{t+1} \tag{11}$$

Plugging equation 11 into equation 10 yields:

$$\tilde{\mathcal{L}}(\theta) = -\mathbb{E}\left[ \omega_{i,j}\log\sigma\left( -\beta\left[ \log\frac{\pi_\theta(\boldsymbol{y}_{t,i}^w|\boldsymbol{y}_{t+1,i}^w)}{\pi_{\text{ref}}(\boldsymbol{y}_{t,i}^w|\boldsymbol{y}_{t+1,i}^w)} - \log\frac{\pi_\theta(\boldsymbol{y}_{t,j}^l|\boldsymbol{y}_{t+1,j}^l)}{\pi_{\text{ref}}(\boldsymbol{y}_{t,j}^l|\boldsymbol{y}_{t+1,j}^l)} \right] \right) \right]$$
$$= -\mathbb{E}\left[ \omega_{i,j}\log\sigma\left( -\beta\left[ \log\pi_\theta(\boldsymbol{y}_{t,i}^w|\boldsymbol{y}_{t+1,i}^w) - \log\pi_{\text{ref}}(\boldsymbol{y}_{t,i}^w|\boldsymbol{y}_{t+1,i}^w) - (\log\pi_\theta(\boldsymbol{y}_{t,j}^l|\boldsymbol{y}_{t+1,j}^l) - \log\pi_{\text{ref}}(\boldsymbol{y}_{t,j}^l|\boldsymbol{y}_{t+1,j}^l)) \right] \right) \right]$$
$$= -\mathbb{E}\left[ \log\sigma\left( -\beta\frac{\beta_{t+1}\alpha_t}{2(1-\bar{\alpha}_t)\alpha_{t+1}}(\|\epsilon_\theta(\boldsymbol{y}_{t+1,i}^w, t) - \epsilon_{t+1}^w + \sigma_{t+1}\epsilon_t^w\|_2^2 - \|\epsilon_{\text{ref}}(\boldsymbol{y}_{t+1,i}^w, t) - \epsilon_{t+1}^w + \sigma_{t+1}\epsilon_t^w\|_2^2 \right. \right.$$
$$\left. \left. - (\|\epsilon_\theta(\boldsymbol{y}_{t+1,j}^l, t) - \epsilon_{t+1}^l + \sigma_{t+1}\epsilon_t^l\|_2^2 - \|\epsilon_{\text{ref}}(\boldsymbol{y}_{t+1,j}^l, t) - \epsilon_{t+1}^l + \sigma_{t+1}\epsilon_t^l\|_2^2)) \right) \right]$$

Similarly, if we approximate $\boldsymbol{y}_t^*$ with the mean of $q\left(\boldsymbol{y}_t^* \mid \boldsymbol{y}_{t+1}^*, \boldsymbol{y}_0^*\right)$, the policy is evaluated as:

$$\pi_\theta(\mathbb{E}\left[\boldsymbol{y}_t^* \mid \boldsymbol{y}_{t+1}^*, \boldsymbol{y}_0^*\right]|\boldsymbol{y}_{t+1}^*) = \frac{1}{\left(\sqrt{2\pi\sigma_{t+1}^2}\right)^d}\exp\left( -\frac{1}{2\sigma_t^2}\left\| \sqrt{\frac{\alpha_t}{\alpha_{t+1}}}(\boldsymbol{y}_{t+1}^* - \frac{\beta_{t+1}}{\sqrt{1-\bar{\alpha}_{t+1}}}\epsilon_{t+1}^*) \right. \right.$$
$$\left. \left. - \sqrt{\frac{\alpha_t}{\alpha_{t+1}}}(\boldsymbol{y}_{t+1}^* - \frac{\beta_{t+1}}{\sqrt{1-\bar{\alpha}_{t+1}}}\epsilon_\theta(\boldsymbol{y}_{t+1}^*, t+1)) \right\|_2^2 \right)$$
$$= \frac{1}{\left(\sqrt{2\pi\sigma_{t+1}^2}\right)^d}\exp\left( -\frac{1}{2\sigma_{t+1}^2}\frac{\alpha_t}{\alpha_{t+1}}\frac{\beta_{t+1}^2}{1-\bar{\alpha}_{t+1}}\|\epsilon_\theta(\boldsymbol{y}_{t+1}^*, t+1) - \epsilon_{t+1}^*\|_2^2 \right)$$
$$= \frac{1}{\left(\sqrt{2\pi\sigma_{t+1}^2}\right)^d}\exp\left( -\frac{1}{2}\frac{1-\bar{\alpha}_{t+1}}{(1-\bar{\alpha}_t)\beta_{t+1}}\frac{\alpha_t}{\alpha_{t+1}}\frac{\beta_{t+1}^2}{1-\bar{\alpha}_{t+1}}\|\epsilon_\theta(\boldsymbol{y}_{t+1}^*, t+1) - \epsilon_{t+1}^*\|_2^2 \right)$$
$$= \frac{1}{\left(\sqrt{2\pi\sigma_{t+1}^2}\right)^d}\exp\left( -\frac{1}{2}\frac{\beta_{t+1}}{(1-\bar{\alpha}_t)}\frac{\alpha_t}{\alpha_{t+1}}\|\epsilon_\theta(\boldsymbol{y}_{t+1}^*, t+1) - \epsilon_{t+1}^*\|_2^2 \right)$$

Again, the log probability is:

$$\log \pi_\theta(\mathbb{E}\left[\boldsymbol{y}_t^* \mid \boldsymbol{y}_{t+1}^*, \boldsymbol{y}_0^*\right]|\boldsymbol{y}_{t+1}^*) = -\frac{1}{2}\frac{\beta_{t+1}\alpha_t}{(1-\bar{\alpha}_t)\alpha_{t+1}}\left\| \epsilon_\theta(\boldsymbol{y}_{t+1}^*, t+1) - \epsilon_{t+1}^* \right\|_2^2$$
$$-\frac{d}{2}\cdot\log 2\pi - d\cdot\log\sigma_{t+1} \tag{12}$$

Plugging equation 12 to equation 10 yields:

$$\hat{\mathcal{L}}(\theta) = -\mathbb{E}\left[ \omega_{i,j}\log\sigma\left( -\beta\left[ \log\frac{\pi_\theta(\mathbb{E}\left[\boldsymbol{y}_{t,i}^w \mid \boldsymbol{y}_{t+1,i}^w, \boldsymbol{y}_{0,i}^w\right]|\boldsymbol{y}_{t,i}^w)}{\pi_{\text{ref}}(\mathbb{E}\left[\boldsymbol{y}_{t,i}^w \mid \boldsymbol{y}_{t+1,i}^w, \boldsymbol{y}_{0,i}^w\right]|\boldsymbol{y}_{t+1,i}^w)} - \log\frac{\pi_\theta(\mathbb{E}\left[\boldsymbol{y}_{t+1,j}^l \mid \boldsymbol{y}_{t+1,j}^l, \boldsymbol{y}_{0,j}^l\right]|\boldsymbol{y}_{t+1,j}^l)}{\pi_{\text{ref}}(\mathbb{E}\left[\boldsymbol{y}_t^l \mid \boldsymbol{y}_{t+1,j}^l, \boldsymbol{y}_{0,j}^l\right]|\boldsymbol{y}_{t+1,j}^l)} \right] \right) \right]$$
$$= -\mathbb{E}\left[ \omega_{i,j}\log\sigma\left( -\beta\left[ \log\pi_\theta(\mathbb{E}\left[\boldsymbol{y}_{t,i}^w \mid \boldsymbol{y}_{t+1,i}^w, \boldsymbol{y}_{0,i}^w\right]|\boldsymbol{y}_{t+1,i}^w) - \log\pi_{\text{ref}}(\mathbb{E}\left[\boldsymbol{y}_{t,i}^w \mid \boldsymbol{y}_{t+1,i}^w, \boldsymbol{y}_{0,i}^w\right]|\boldsymbol{y}_{t+1,i}^w) \right. \right. \right.$$
$$\left. \left. \left. - (\log\pi_\theta(\mathbb{E}\left[\boldsymbol{y}_{t,j}^l \mid \boldsymbol{y}_{t+1,j}^l, \boldsymbol{y}_{0,j}^l\right]|\boldsymbol{y}_{t+1,j}^l) - \log\pi_{\text{ref}}(\mathbb{E}\left[\boldsymbol{y}_t^l \mid \boldsymbol{y}_{t+1,j}^l, \boldsymbol{y}_{0,j}^l\right]|\boldsymbol{y}_{t+1,j}^l)) \right] \right) \right]$$
$$= -\mathbb{E}\left[ \log\sigma\left( -\beta\frac{\beta_{t+1}\alpha_t}{2(1-\bar{\alpha}_t)\alpha_{t+1}}(\|\epsilon_\theta(\boldsymbol{y}_{t+1,i}^w, t+1) - \epsilon_{t+1}^w\|_2^2 - \|\epsilon_{\text{ref}}(\boldsymbol{y}_{t+1,i}^w, t+1) - \epsilon_{t+1}^w\|_2^2 \right. \right.$$
$$\left. \left. - (\|\epsilon_\theta(\boldsymbol{y}_{t+1,j}^l, t+1) - \epsilon_{t+1}^l\|_2^2 - \|\epsilon_{\text{ref}}(\boldsymbol{y}_{t+1,j}^l, t+1) - \epsilon_{t+1}^l\|_2^2)) \right) \right]$$

# B RELATED WORKS

## B.1 DIFFUSION BASED TEXT-TO-IMAGE MODELS

Diffusion models (Ho et al., 2020; Sohl-Dickstein et al., 2015; Song et al., 2020; Dhariwal & Nichol, 2021) have been the state-of-the-art in image generation. A diffusion model consists of a forward process that gradually adds noise to the image and attempts to learn to reverse this process with a neural network. Diffusion-based T2I models have also achieved impressive results in producing high-quality images that closely adhere to the given caption (Nichol et al., 2022; ram, 2022; Rombach et al., 2022b; Saharia et al., 2022; Podell et al., 2023a). However, these models are pre-trained on large web dataset and may not align with human preference. Our work aims to tackle this issue by improving T2I model alignment using semantically related prompt-image pairs

## B.2 DIFFUSION-DPO

Diffusion-DPO (Wallace et al., 2024) generalizes the efficient DPO (Rafailov et al., 2023) to diffusion model alignment. Diffusion-DPO defines reward $R(\boldsymbol{x}, \boldsymbol{y}_{0:T})$ on the diffusion path for prompt $\boldsymbol{x}$, latents $\boldsymbol{y}_{1:T}$ and image $\boldsymbol{y}_0$. The RLHF loss is defined as:

$$\max_{p_\theta} \mathbb{E}_{\boldsymbol{x} \sim \mathcal{D}_x, \boldsymbol{y}_{0:T} \sim p_\theta(\boldsymbol{y}_{0:T}|\boldsymbol{x})} \left[ r\left(\boldsymbol{x}, \boldsymbol{y}_0\right) \right] - \beta \mathbb{D}_{\mathrm{KL}} \left[ p_\theta \left(\boldsymbol{y}_{0:T} \mid \boldsymbol{x}\right) \| p_{\mathrm{ref}} \left(\boldsymbol{y}_{0:T} \mid \boldsymbol{x}\right) \right] \quad (13)$$

where $r\left(\boldsymbol{x}, \boldsymbol{y}_0\right) = \mathbb{E}_{p_\theta(\boldsymbol{y}_{1:T}|\boldsymbol{y}_0, \boldsymbol{x})} \left[ R\left(\boldsymbol{x}, \boldsymbol{y}_{0:T}\right) \right]$ is the marginalized image reward across all possible diffusion trajectories.

With some approximation and algebra, the Diffusion-DPO loss can be simplified to:

$$\begin{aligned} L(\theta) = -\mathbb{E}_{\substack{(\boldsymbol{y}_0^w, \boldsymbol{y}_0^l) \sim \mathcal{D}, t \sim \mathcal{U}(0,T), \\ \boldsymbol{y}_t^w \sim q(\boldsymbol{y}_t^w|\boldsymbol{y}_0^w), \\ \boldsymbol{y}_t^l \sim q(\boldsymbol{y}_t^l|\boldsymbol{y}_0^l)}} \log \sigma \bigg( &- \beta T \omega\left(\lambda_t\right) \left(\|\boldsymbol{\epsilon}^w - \boldsymbol{\epsilon}_\theta\left(\boldsymbol{y}_t^w, t\right)\|_2^2 - \|\boldsymbol{\epsilon}^w - \boldsymbol{\epsilon}_{\mathrm{ref}}\left(\boldsymbol{y}_t^w, t\right)\|_2^2 \right. \\ &- \left( \left\|\boldsymbol{\epsilon}^l - \boldsymbol{\epsilon}_\theta\left(\boldsymbol{y}_t^l, t\right)\right\|_2^2 - \left\|\boldsymbol{\epsilon}^l - \boldsymbol{\epsilon}_{\mathrm{ref}}\left(\boldsymbol{y}_t^l, t\right)\right\|_2^2 \right) \bigg) \end{aligned}$$

$$(14)$$

The approximate Diff-contrast loss is a function of denoising diffusion loss, which is simple to compute. However, it is limited to the prompt-image pairs that shares identical prompts.

## B.3 RELATIVE PREFERENCE OPTIMIZATION

Relative Preference Optimization (Yin et al., 2024) (RPO) draws inspiration from human cognition that often involves interpreting divergent responses, not only to identical questions but also to those that are similar. RPO utilizes both identical and semantically related prompt-response pairs in a batch and weight them by the similarity of the prompts between preferred and rejected data. Assume we have unpaired data with N preferred prompt-response pairs and M rejected prompt-response pairs, $(y_{w,i}, x_i), (y_{l,j}, x_j), i \in [N], j \in [M]$. The RPO loss function is given by:

$$\mathcal{L}_{RPO} = -\frac{1}{M * N} \sum_{i=1}^{M} \sum_{j=1}^{N} \log \sigma \left( \omega_{ij} \times \beta \left( \log \frac{\pi_\theta\left(y_{w,i} \mid x_i\right)}{\pi_{ref}\left(y_{w,i} \mid x_i\right)} - \log \frac{\pi_\theta\left(y_{l,j} \mid x_j\right)}{\pi_{ref}\left(y_{l,j} \mid x_j\right)} \right) \right) \quad (15)$$

where

$$\omega_{ij} = \frac{\tilde{\omega}_{ij}}{\sum_{j'=1}^{N} \tilde{\omega}_{ij'}}, \quad \tilde{\omega}_{ij} = \exp\left( -\frac{\cos\left(f\left(x_{w,i}\right), f\left(x_{l,j}\right)\right)}{\tau} \right)$$

where $f(\cdot)$ is a text encoder, and $\tau$ is the temperature parameter. Intuitively, RPO constructs a correlation matrix of prompts within the mini-batch and normalizes the weights across each column to incorporate semantically related prompts into preference optimization.

RPO significantly improves the alignment of LLMs with human preferences by better emulating the human learning process through innovative contrastive learning strategies.

Table 3: Ablation Study on Diff-contrast Temperature Parameters. We use SD1.5 as the base model for Huamn preference alginment and SFT-tuned SD1.5 for Van Gogh Style alignment. We reported reward model scores on HPSV2 test prompts and style alignment FID under various temperatures, the $beta$ parameter is set to be 2000.

| Temperature Parameter ($\tau$) | 0.01 | 0.1 | 1.0 | 2.0 | 5.0 |
|---|---|---|---|---|---|
| HPS | **27.373** | 27.286 | 27.245 | 27.176 | 27.205 |
| Pick Score | 21.425 | **21.448** | 21.417 | 21.366 | 21.421 |
| Aesthetics | 5.694 | 5.696 | 5.671 | **5.733** | 5.657 |
| CLIP | 0.351 | 0.352 | **0.353** | 0.353 | 0.352 |
| Image Reward | **0.342** | 0.335 | 0.320 | 0.270 | 0.316 |
| Style Alignment FID | 31.96 | 16.26 | 13.66 | 13.82 | **13.25** |

## C  ADDITIONAL ABLATION STUDIES

n this section, we begin by presenting the ablation studies on the temperature parameter $\tau$. We also conducted ablation studies on batch size and $\beta$ in appendix C. In the batch size ablation study, we increased the batch size from 2 to 16 and observed a consistent improvement in HPSv2 scores. This upward trend indicates that Diff-contrast's approach efficiently leverages larger batches of comparative data to enhance preference learning. This trend demonstrates how effectively Diff-contrast's strategy utilizes larger quantities of comparative data for preference learning. For ablation studies on $\beta$, we explored values from the set [2000, 5000, 7000, 10000] and observed that $\beta = 2000$ yielding the best performance.

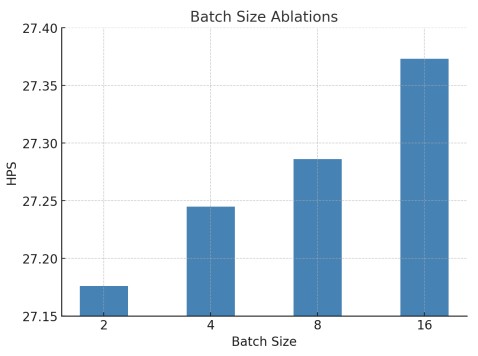

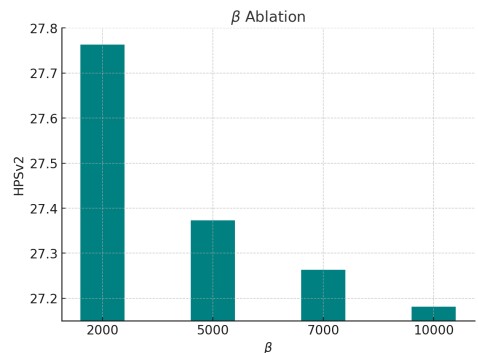

(a) Ablation Studies with respect to Batch Size    (b) Ablation Studies with respect to $\beta$

Figure 5: Ablation Studies based on Diff-contrast-SD1.5

## D  ADDITIONAL DETAILS:

We used $\beta = 2000$ for SD1.5 human preference alignment and $\beta = 5000$ for the rest of the experiments. Following Wallace et al. (2024), we use AdamW for SD 1.5 (Loshchilov & Hutter, 2019) and AdaFactor for SDXL (Shazeer & Stern, 2018). For human preference alignment, we adopted a learning rate of $\frac{2000}{\beta}2.048 \cdot 10^{-8}$ for Diff-contrast. For style alignment, the learning rate is set to be $\frac{2000}{\beta}2.048 \cdot 10^{-8}$ in the first stage. In second stage, we use $\frac{2000}{\beta}2.048 \cdot 10^{-8}$ for Van Gogh and $\frac{2000}{\beta}2.048 \cdot 10^{-9}$ for Sketch and Winter datasets. Eight Nvidia A100 GPUs were used for training. The local batch size on each GPU is 16 with gradient accumulation of 4 steps for SD 1.5. For SDXL, the local batch size is 8 with gradient accumulation of 8 steps.

## E    SDXL Style Alignment Details

Table 4: Additional details for SDXL style alignment experiments.

|  | SFT LR | SFT Steps | Second Stage LR | Second Stage Steps |
|---|---|---|---|---|
| Van Gogh | 1e-7 | 2000 | 1e-8 | 1000 |
| Sketch | 1e-7 | 2000 | 1e-8 | 500 |
| Winter | 1e-7 | 2000 | 1e-8 | 1000 |

## F    Prompts and More Human Preference Alignment Examples

We list the prompts used in Figure 1 as follows

1. Kyoto Animation stylized anime mixed with tradition Chinese artworks  A dragon flying at modern cyberpunk fantasy world. Cinematic Lighting, ethereal light, gorgeous, glamorous, 8k, super detail, gorgeous light and shadow, detailed decoration, detailed lines

2. a masterpiece, batgirl/supergirl from dc comic wearing alternate yellow costume, coy and alluring, full body, Kim Jung gi, freedom, soul, digital illustration, comic style, cyberpunk, perfect anatomy, centered, approaching perfection, dynamic, highly detailed, watercolor painting, artstation, concept art, smooth, sharp focus, illustration, art by Carne Griffiths and Wadim Kashin, unique, award winning, masterpiece

3. (Pirate ship sailing into a bioluminescence sea with a galaxy in the sky) , epic, 4k, ultra,

4. In the vast emptiness of the interstellar void, two omnipotent deities from separate dimensions collide. The Creator, wields the power of creation, bringing forth stars and life forms with a wave of their hand.  Opposite them stands the Destroyer, a god whose face is a shifting visage of decay and entropy disintegrating celestial bodies and extinguishing stars.

5. neon light art, in the dark of night, moonlit seas, clouds, moon, stars, colorful, detailed, 4k, ultra hd, realistic, vivid colors, highly detailed, UHD drawing, pen and ink, perfect composition, beautiful detailed intricate insanely detailed octane render trending on artstation, 8k artistic photography, photorealistic concept art, soft natural volumetric cinematic perfect light

6. Lady with floral headdress, blonde hair, blue eyes, seductive attire, garden, sunny day, realistic texture

7. a cute little matte low poly isometric Zelda Breath of the wild forest island, waterfalls, mist, lat lighting, soft shadows, trending on artstation, 3d render, monument valley, fez video game

8. a psychedelic surrealist painting of a sunflower, surreal, dripping, melting, Salvador Dali, Pablo Piccaso

9. Commercial photography, powerful explosion of golden dust, luxury parfum bottle, yellow sunray, studio light, high resolution photography, insanely detailed, fine details, isolated plain, stock photo, professional color grading, 8k octane rendering, golden blury background

10. Digital painting of a beautiful young Japanese woman, dancing, kimono is crafted with waves and layers of fabric, creating a sense of movement and depth, fall season, by (random: famous japanese artists) , artstation, 8k, extremely detailed, ornate, cinematic lighting, rim lighting,

11. giraffe in flowers by artist arne thun, in the style of natalia rak, 8k resolution, vintage aesthetics, wallpaper, animated gifs, naoto hattori, highly realistic

12. bright painting of a tree with stars in the sky, in the style of dreamlike fantasy creatures, multilayered dimensions, swirling vortexes, realistic color schemes, dark green and light blue, light-filled landscapes, mystic mechanisms

13. Holy male paladin, looking straight at the viewer, serious facial expression, heavenly aura, bokeh, light bloom, bright light background, cinematic lighting, depth of field, concept art, HDR, ultra high-detail, photorealistic, high saturation

14. poster of a blue dragon with sunglasses, in the style of orient-inspired, post-apocalyptic surrealism, light orange and red, 32k uhd, chinapunk, meticulous design, detailed costumes

15. Double Exposure Human-Nature, trees, flowers, mountain, sunset, nature mind, expressive creative art, surrealistic concept art, ethereal landscape in a cloud of magic coming out the top of a human head, incredible details, high-quality, flawless composition, masterpiece, highly detailed, photorealistic, 8k sharp focus quality surroundings

Additional sample images from Diff-contrast-SDXL are presented in Figure 6 and Figure 7.

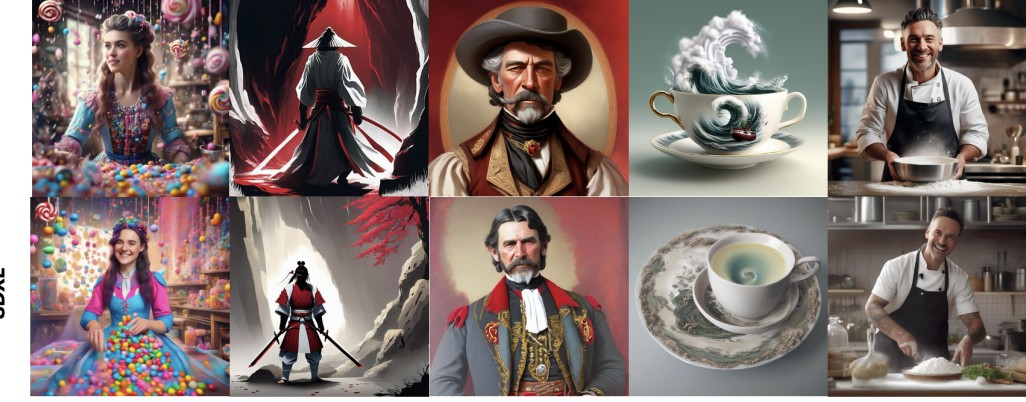

Figure 6: **Sample images from Diff-contrast-SDXL** The prompts used to generate the images are: "A whimsical candy maker in her enchanted workshop, surrounded by a cascade of multicolored candies falling like rain, wearing a bright, patchwork dress, her hair tinted with streaks of pink and blue, 8K, hyper-realistic, cinematic, post-production.","A samurai cloaked in white with swords stands in a light beam of a dark cave, with a ruby red sorrow evident in the image.","Victorian genre painting portrait of Royal Dano, an old west character in fantasy costume, against a red background.", "A typhoon in a tea cup, digital render","A charismatic chef in a bustling kitchen, his apron dusted with flour, smiling as he presents a beau- tifully prepared dish. 8K, hyper-realistic, cinematic, post-production."

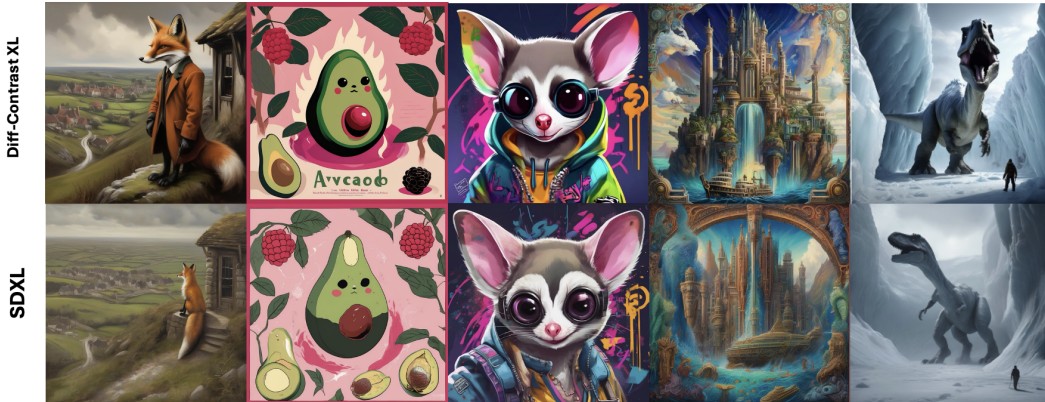

Figure 7: **Sample images from Diff-contrast-SDXL** The prompts used to generate the images are: "An oil painting of an anthropomorphic fox overlooking a village in the moor.","A graphic poster featuring an avocado and raspberry observing a burning world, inspired by old botanical illustrations, Matisse, Caravaggio, Basquiat, and Japanese art." ,"A colorful anime painting of a sugar glider with a hiphop graffiti theme, by several artists, currently trending on Artstation.", "A detailed painting of Atlantis by multiple artists, featuring intricate detailing and vibrant colors.", "A giant dinosaur frozen into a glacier and recently discovered by scientists, cinematic still"

## G   MORE STYLE ALIGNMENT EXAMPLES

Additional sample images from style-aligned Stable Diffusion models are presented in Figure 8.

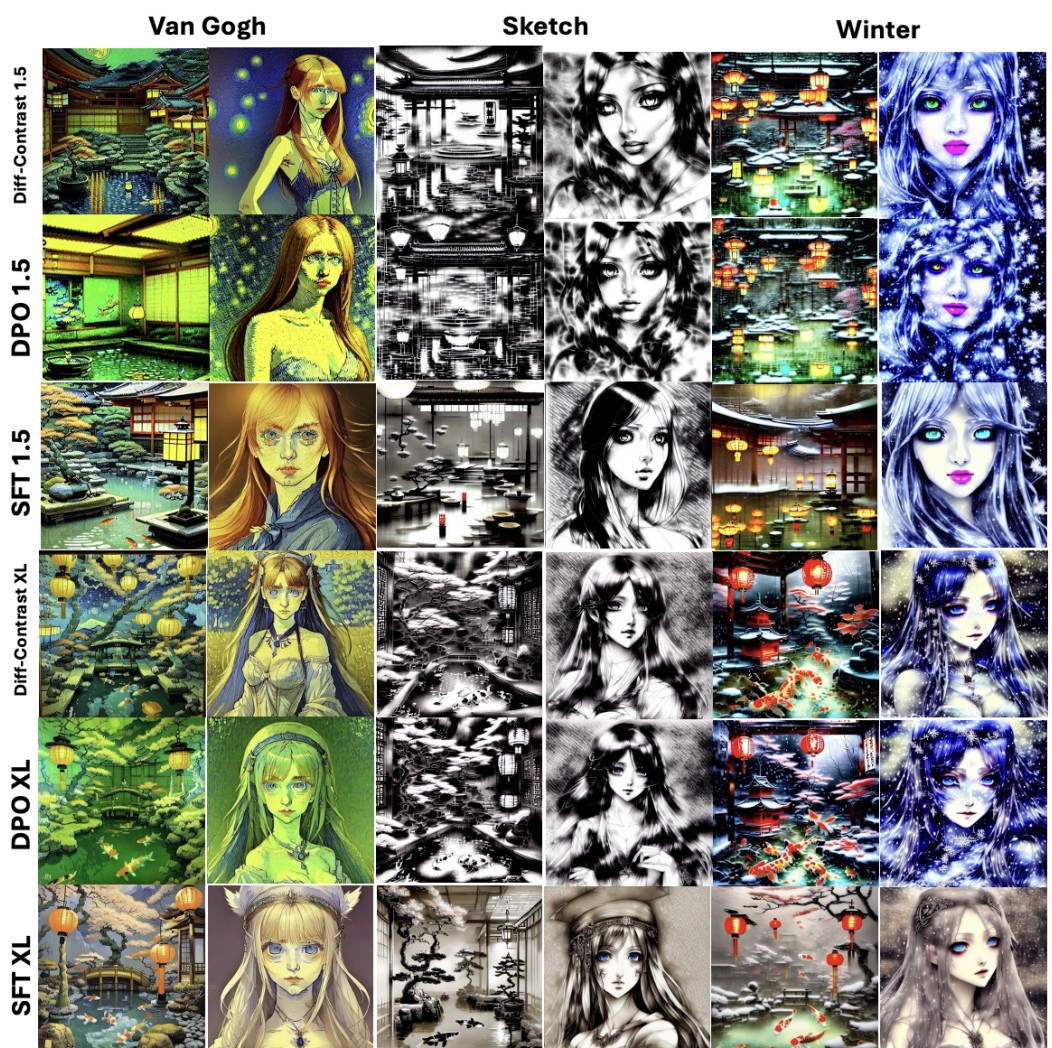

Figure 8: **Sample images from Style Alignment Task** , the images are generated from prompts: "Japanese hot spring interior with lanterns, koi fish and bonsai trees in a painting by Greg Rutkowski and Craig Mullins.","A beautiful girl posing dramatically, with stunning eyes and features, by Davinci on Pixiv."

## H SOCIAL IMPACTS

Diff-contrast enhances the performance of diffusion based T2I models that could democratize artistic creation, allowing individuals without formal training in art to generate high-quality images. This can inspire creativity and make art more accessible. However, there's a risk of these models being used to create misleading images or deepfakes, potentially spreading misinformation or harming individuals' reputations.

