# OpenReview forum: "Diffusion Preference Alignment via Relative Text-Image Contrast"
_ICLR.cc/2025/Conference — Submitted to ICLR 2025_

### Official Review · Reviewer_4f3P · 2024-11-03

**Soundness:** 3
**Presentation:** 2
**Contribution:** 3
**Rating:** 6
**Confidence:** 3

**Summary:**

This paper aims to align diffusion models with human preferences by optimizing relative preferences. Previous work uses a reward model to optimize the diffusion model but might suffer from high computational cost and fail to accurately capture human preferences. This paper proposes Diff-contrast loss along with contrastive weighting to enable the diffusion models to learn to align the preference effectively from contrasting images generated from different prompts. Moreover, this paper contributed a task named Style Alignment along with a dataset to aid in the evaluation of preference alignment methods without human annotators. The experiment results indicated its effectiveness with performance compared to other baselines.

**Strengths:**

1. The paper is well-structured overall and easy to follow with good clarity in writing.
2. This paper presents a new method to utilize the information in the prompt pairs for preference alignments for diffusion models and the originality is sufficient.
3. The significance of this paper is good with the aim of helping diffusion models generate images that align better with human preference in terms of image details or overall expression.
4. The qualitative comparison of the proposed method and other baselines shows the effectiveness of this method with better image quality.

**Weaknesses:**

1. The paper contributed a dataset of three styles for the evaluation of style alignment, however, the generated results are evaluated with FID scores that might not fully capture the information of the styles.
2. There are several minor writing suggestions:

- Page 2 line 77 "Diffusion-DPO (Wallace et al., 2024) and incorporates Direct Preference Optimization (DPO) (Rafailov et al., 2023) into"

- The notation of the formulas needs to be further clarified or rearranged. Such as moving the definition of preference data $(y^w, y^l, x)$ or the notation of $w, l$ before the equations (1) and (2) to explain what $w, l$ are in the equations; $\omega_{i, j}$ in equation (2) refers to the contrastive weight but was not indicated until section 2.1.

- The bond fonts in Table 1 are not clear in meaning.

- A brief discussion on the metrics in 4.1 would be helpful.
3. See the questions below.

**Questions:**

1. The styles (Van Goph, Sketch, and Winter) do not seem to be parallel. How are the styles for the style alignment task selected? Is there any specific concern when the dataset for style alignment is constructed?
2. The images in the datasets are generated with image editing models, and how do you guarantee the generated images fit human preference in terms of styles? If not, how do the authors make sure the style alignment task is helpful for the evaluation of preference alignment?
3. For the two-stage fine-tuning experiments for style alignment, is there any specific reason to use SFT along with SFT/DPO/Contrast instead of using the same methods twice?
4. In abstract lines 50-51, the authors mentioned that "both prompt-image pairs with identical prompts and those with semantically related content across various modalities", what does the prompt-image pairs with semantically related content mean? Does it refer to the contrastive weight in section 2.1?

---

> ### Author Response · Authors · 2024-11-22
>
> We thank reviewer 4f3P for your positive feedback and constructive suggestions, we will address your concerns point by point:
>
> * **W1**: It is generally hard to train a reward model that fully capture the information of the styles, just like the human preference models may not fully reflect human preference. Evaluation with FID strikes a balance between effectiveness and computation costs.
>
> * **W2**: Thanks for pointing out the typos, we will fix them in an updated version.
>
> *  For **Q1** and **Q2**. We selected styles that maximize sample efficiency, as this process requires human evaluation of style transfer success. These three styles were chosen because they are relatively easy to transfer to most pick-a-pic subset images, have strong distribution shifts, and thus serve as appropriate tasks. The images were selected based on their correct style representation, as human preference is not the priority in this task.
>
> * **Q3**: there is a distribution shift between the pre-trained data and our style-transferred data. It is standard practice to use SFT to close the distribution gap when the alignment goal is different from the pre-trained model, as mentioned in [1].
>
> [1] Bram Wallace, Meihua Dang, Senthil Purushwalkam, Rafael Raifalov, Stefano Ermon, Linqi Zhou, Aaron Lou, Caiming Xiong, Nikhil Naik, Shafiq Joty, Diffusion Model Alignment Using Direct Preference Optimization, 2024
>
> * **Q4**: Yes, the contrastive weighting scheme assigns higher weights to the pairs with higher similarity, the “prompt-image pairs with semantically related content mean” are prioritized in this scheme. Consequently, prompt-image pairs with identical prompts and those with semantically related content will be utilized.

---

> > ### Comment · Reviewer_4f3P · 2024-11-25
> >
> > Thanks for the authors' reply. I have the following concerns after reading this reply: (1) since the datasets with the given styles are not related to human preference explicitly, how does the dataset work in terms of evaluating how well the methods align with human preference? The examples in Fig 1 and Fig 2 seem to show more complex and subtle control over the image styles. (2) I'm not fully convinced with the evaluation with only FID score. As Reviewer DP5H mentioned, could the gram matrix distance or style loss work better for this task?

---

> > > ### Author Response · Authors · 2024-11-25
> > >
> > > We thank Reviewer 4f3P fo your constructive feedbacks:
> > >
> > > * The datasets are not constructed to reflect human preference. The goal of style alignment is to evaluate how well can the Alignment Algorithm shift the model distribution towards the preferred distribution, which includes, but not limited to human preference alignment. The motivation of this task is explained in Section 3.
> > >
> > > * We acknowledge the effectiveness of Gram Matrix distance in style transfer tasks. While FID is commonly used in generative modeling to measure statistical distances between image sets, both metrics are valid depending on perspective - FID for generative modeling and Gram Matrix distance for style transfer. We appreciate the reviewer's suggestion and will incorporate discussion of Gram Matrix distance in our revision.

---

> > > > ### Comment · Reviewer_4f3P · 2024-11-25
> > > >
> > > > Thanks for the reply from the authors. It makes sense to quantitatively evaluate the performance of style transfer with simpler settings on the constructed datasets but it would be better to discuss more regarding the transition to the application since the application scenario is aimed at more subtle styles for human preference alignment. Besides this, my overall concerns have been addressed and I would like to keep my original rating.

---

### Official Review · Reviewer_TQH8 · 2024-11-03

**Soundness:** 3
**Presentation:** 2
**Contribution:** 2
**Rating:** 5
**Confidence:** 4

**Summary:**

In this paper, the authors introduce the Diff-contrast model combining the diffusion model with the RPO strategy. A novel loss function and style alignment dataset are proposed. In the style alignment task, experiments prove the advantage of the proposed Diff-contrast compared with SDXL model, SFT-SDXL and DPO-SDXL.

**Strengths:**

This paper continues the previous research on human preference in the Text-to-Image area, and constructs a new loss function by introducing a relative strategy from LLM area. At the same time, in order to successfully construct the loss, a corresponding reference dataset is constructed, and the effectiveness of the method is verified by comparative experiments.

**Weaknesses:**

1. Similar to the Diffusion-DPO approach, this article is more like a simple change of strategy. There is no obvious innovation.
2. Personal preference is achieved in T2I primarily through textual descriptions. This paper argues that the task of reconstruction is mainly through the consistency of the generated samples and reference datasets. However, the real massive data samples have been trained and integrated into the pre-trained model during the training phase.

**Questions:**

1.  More comparisons with LoRA methods need to be verified. Since the style change of images in the T2I task can be achieved by way of LoRa training.
2. Images in the Figure 1 need to compare with the other generation of similar style images.
3. It is necessary to compare this strategy with other T2I models, like Flux 1.0 and SD3, to show its superiority. Rather than just comparing it to the base model.
4. The generated results in each figure should be displayed with quantitative evaluation indicators to demonstrate their superiority.

---

> ### Author Response · Authors · 2024-11-22
>
> We appreciate the comments from  reviewer TQH8, we will address your concerns point by point:
>
> * **W1**: We acknowledge the level of novelty may be perceived differently, yet, we believe generalizing RPO to Diffusion models is a significant contribution: as we suggest in Section 2, the multi-step sampling process of Diffusion Models is way more complicated than the next token prediction in LLM. We proposed an algorithm that aligns diffusion models at each step of the diffusion chain. Additionally, we designed contrastive weights for more complicated multi-modal data. It requires decent innovation to generalize a preference optimization algorithm from LLMs to T2I models.
>
> * **W2**: We agree that the evaluation of style alignment was done on training data since there’s no existing metric to evaluate how well the model adapts to a certain style. In addition, our figures 4 and 8 utilizes out-of-domain prompts and the model performed well. We have considered picking an equal-sized test set to further validate the effectiveness, however, picking prompt-image pairs that fits the desired styles is too time consuming.
>
> * **Q1**: Figure 1 is an example of our generation, please checkout Figure 3 for comparison with other algorithms
>
> * **Q2**: Our preliminary experiments suggest that LoRA tuning also works, however, reproducing all experiments on SD1.5 and SDXL even with LoRA is very time consuming, we will leave this for future exploration.
>
> * **Q3**: To evaluate the performance of an alignment algorithm, it is best to control all variables and focus on the algorithm itself. Stable Diffusion 1.5 and Stable Diffusion XL are the most popular open-source text-to-image models in medium and large scales, respectively. The experiments on these two models sufficiently demonstrate the effectiveness and generalizability of our algorithm. Adapting our algorithm to Flux1.0 and SD3 requires considerable effort, and we will explore this in future work.
>
> * **Q4**: We have conducted quantitative evaluation in Table 1. The purpose of the example figures are for qualitative evaluation.

---

### Official Review · Reviewer_DP5H · 2024-11-07

**Soundness:** 1
**Presentation:** 2
**Contribution:** 1
**Rating:** 3
**Confidence:** 3

**Summary:**

In this paper, the authors propose a new diffusion alignment method based on RPO. They also provide a newly selected subset of the Pick-a-Pic V2 dataset for style transfer. They conduct experiments in comparison with baselines including Diffusion-DPO and observe higher win rates.

**Strengths:**

This paper aims to tackle an important and interesting problem, text-to-image diffusion alignment. Qualitatively, the proposed method seems to deliver good results for the tasks experimented in this paper.

**Weaknesses:**

My main concern about this paper lies in its novelty.
1. The first major contribution of this paper is a simple adaptation of RPO to diffusion models. While it seems to provide decent performance, the technical novelty is not very significant.
2. The second major contribution is the so-called “new” task “style alignment”. This task is identical to the well-established task style transfer. In fact in the definition the authors provide, they describe it as “Style alignment aims to fine-tune T2I models to generate images that align with the offline data to achieve style transfer on T2I models” (Line 320-321). Just rebranding it as “alignment” does not make it an actual novel task.

Besides novelty, I also have the following concerns regarding the evaluations conducted in this paper:

3. Style transfer task has a well-established evaluation metric, i.e. gram matrix distance. While FID can also tell part of the story, adding the standard metric can be better.

4. In the human preference evaluation, the authors only provide the win rate as their comparison results (they claim that they have provided the average scores in Line 378, however, I personally could not find it). Win rate cannot provide the information of the improvement margin gained by the proposed method. In other words, it is possible that the proposed method wins 70% of the time but still has a lower average score when all the winning cases are won by small margin and all the losing cases are lost by large margin. I think providing the raw scores is essential for this comparison.


Finally, I would like to make some suggestions about the general writing of this paper:

5. This paper is also structured in a way that mixes prior work with their contributions. I would suggest a clearer cut between the literature and the new contribution by including proper related works and background sections in the main paper if the authors ever consider restructuring the paper.

**Questions:**

It would be great if the authors can address the weakness mentioned above.
In addition, I am wondering if the authors can provide additional information about how they define the multimodal embedding function $f$.

---

> ### Author Response · Authors · 2024-11-22
>
> We thank reviewer DP5H for your valuable feedbacks, we will address your concerns point by point:
> *  **W1**: We acknowledge that the level of novelty may be perceived differently; however, we believe that generalizing RPO to diffusion models is a significant contribution. As we discussed in Section 2, the multi-step sampling process of diffusion models is far more complicated than the next-token prediction in LLMs. We have proposed an algorithm that aligns diffusion models at each step of the diffusion chain. Additionally, we have designed contrastive weights for more complex multi-modal data. It requires significant innovation to generalize a preference optimization algorithm from LLMs to T2I models.
> * **W2**: Yes, the style alignment task is motivated by and based on style transfer, which is mentioned in section 3. Our motivation is twofold: 1) to measure the performance of Direct Alignment Algorithms under preference distributions where winner and loser clusters have strong distribution shifts, and 2) to provide an economic metric for evaluating image preference learning methods that distinguishes between these methods more effectively. We believe that "re-branding" would twist our presentation, as we have already acknowledged our aim to achieve style transfer via alignment algorithms in line 321 and cited relevant literature. In this regard, the name "style alignment" articulates our goal clearly by specifying the methodological context.
>
> * **W3**: We appreciate this constructive feedback, we agree that FID is an effective evaluation metric, we will consider adding the gram matrix into our next version due to time limit.
> * **W5**: Again, We acknowledge that the level of novelty may be perceived differently. However, we will continue to improve the clarity of our presentation.
> * **Q1**: The $f(x_{\text{prompt}}, y_{\text{image}})$ function is the clip encoder from  openai/clip-vit-large-patch14

---

> > ### Author Response · Authors · 2024-11-23
> >
> > * **Q4**:
> > Please find the table attached:
> >
> > | Model  | Test Dataset   | Method                     |    HPS    | Pick Score | Aesthetics |  CLIP   | Image Reward |
> > |--------|----------------|----------------------------|-----------|------------|------------|---------|--------------|
> > | **SD1.5**  | **HPSV2**        | **Base**                     | 26.931    | 20.948     | 5.554      | 0.345   | 0.083        |
> > |          |                | SFT                        | 27.533    | 21.499     | 5.810      | 0.349   | 0.553        |
> > |          |                | DPO                        | 27.233    | 21.396     | 5.679      | 0.348   | 0.296        |
> > |          |                | Contrast (ours)            | **27.755**| **21.847** | **5.803**  | **0.352**| **0.620**    |
> > |          | **Parti-Prompt** | **Base**                   | 26.704    | 21.375     | 5.330      | 0.332   | 0.159        |
> > |          |                | SFT                        | 27.105    | 21.681     | 5.464      | 0.332   | 0.552        |
> > |          |                | DPO                        | 26.934    | 21.626     | 5.462      | 0.327   | 0.321        |
> > |          |                | Contrast (ours)            | **27.388**| **21.993** | **5.457**  | **0.337**| **0.587**    |
> > | **SDXL**   | **HPSV2**        | **Base**                     | 27.900    | 22.694     | 6.135      | 0.375   | 0.808        |
> > |          |                | SFT                        | 27.555    | 22.069     | 5.847      | 0.373   | 0.692        |
> > |          |                | DPO [Wallace et al., 2024] | 28.082    | 23.159     | 6.139      | 0.383   | 1.049        |
> > |          |                | Contrast (ours)            | **28.658**| **23.208** | **6.163**  | **0.383**| **1.046**    |
> > |          | **Parti-Prompt** | **Base**                   | 27.721    | 22.562     | 5.766      | 0.357   | 0.668        |
> > |          |                | SFT                        | 27.081    | 21.657     | 5.536      | 0.358   | 0.545        |
> > |          |                | DPO                        | 27.637    | 22.894     | 5.806      | **0.370**| 1.032        |
> > |          |                | Contrast (ours)            | **28.456**| **22.980** | **5.872**  | 0.367   | **1.077**    |

---

> > > ### Comment · Reviewer_DP5H · 2024-11-25
> > > **Thank you for your rebuttal response**
> > >
> > > Thank you for your rebuttal response. I appreciate that the authors answer my forth weakness concern. However, evident from their response, it seems like the authors and I disagree significantly on the novelty aspect of this paper. Since the reasoning in their response cannot convince me, and they have not provided any revision of the paper like they have mentioned in their answer to W5, I would like to keep my original score.

---

### Meta-Review · Area_Chair_iXk4 · 2024-12-17

**Metareview:**

[Summary]
The paper proposes Diff-contrast, a method for aligning diffusion models with human preferences using a new loss function and contrastive weighting. A "Style Alignment" task and a new dataset are introduced to evaluate preference alignment without human annotators. Experiments show improved performance compared to baselines like Diffusion-DPO and SDXL.

[Strengths]
1. The paper addresses the important problem of aligning diffusion models with human preferences.
2. The Diff-contrast loss function and Style Alignment task provide new tools for evaluating preference alignment.
3. The paper is well-written and easy to follow.

[Weaknesses]
1. Limited Novelty: The approach is incremental, and the style alignment task essentially rebrands style transfer.
2. Evaluation Issues: FID scores may not capture style details, and win rates lack clarity about the margin of improvement. Raw scores would improve evaluation.
3. Limited Comparisons: The paper does not compare with other state-of-the-art methods like LoRA and Flux 1.0.
4. Dataset and Task Concerns: Style selection is not well justified, and using image editing models raises concerns about human preference alignment.

[Decision]
The paper presents an interesting approach but lacks significant novelty and thorough evaluation. Based on recommendations from reviewers (3: Reject, 5: Marginally below the threshold, 6: Marginally above the threshold), I recommend rejecting this paper.

**Additional Comments On Reviewer Discussion:**

During the rebuttal period, Reviewer DP5H raised concerns about the novelty of the paper, particularly regarding the adaptation of RPO and the rebranding of style transfer as "style alignment." DP5H also pointed out issues with the evaluation metrics, such as the reliance on FID scores and the lack of raw scores in human preference comparisons. Reviewer TQH8 shared similar concerns about the incremental nature of the work and requested more comparisons with models like LoRA and Flux 1.0. Reviewer 4f3P focused on the dataset construction and questioned how well it reflects human preferences, asking for further clarification.
While the authors provided some responses, they did not fully address these major concerns. The novelty and evaluation issues remained unresolved, and no significant revisions were made. These points contributed to the decision to recommend rejection.

---

### Decision · Program_Chairs · 2025-01-22

Reject